# MLZero: A Multi-Agent System for End-to-end Machine Learning Automation

**Haoyang Fang, Boran Han, Nick Erickson, Xiyuan Zhang, Su Zhou, Anirudh Dagar, Jiani Zhang, Ali Caner Turkmen, Cuixiong Hu, Huzefa Rangwala, Ying Nian Wu, Bernie Wang, George Karypis**

**Amazon Web Services**

{haoyfang,boranhan,neerick,xiyuanz,suzhou,anidagar}@amazon.com
{zhajiani,atturkm,tonyhu,rhuzefa,wunyin,yuyawang,gkarypis}@amazon.com

🌐 **Website**: https://project-mlzero.github.io/
⬡ **GitHub**: https://github.com/autogluon/autogluon-assistant

## Abstract

Existing AutoML systems have advanced the automation of machine learning (ML); however, they still require substantial manual configuration and expert input, particularly when handling multimodal data. We introduce MLZero, a novel multi-agent framework powered by Large Language Models (LLMs) that enables end-to-end ML automation across diverse data modalities with minimal human intervention. A cognitive perception module is first employed, transforming raw multimodal inputs into perceptual context that effectively guides the subsequent workflow. To address key limitations of LLMs, such as hallucinated code generation and outdated API knowledge, we enhance the iterative code generation process with semantic and episodic memory. MLZero demonstrates superior performance on MLE-Bench Lite, outperforming all competitors in both success rate and solution quality, securing six gold medals. Additionally, when evaluated on our Multimodal AutoML Agent Benchmark, which includes 25 more challenging tasks spanning diverse data modalities, MLZero outperforms the competing methods by a large margin with a success rate of 0.92 (+263.6%) and an average rank of 2.28. Our approach maintains its robust effectiveness even with a compact 8B LLM, outperforming full-size systems from existing solutions.

## 1 Introduction

The quest to democratize machine learning has long been championed by AutoML, a field dedicated to automating the intricate process of building ML solutions [21, 20, 19, 67, 65, 28]. However, conventional AutoML systems often struggle to manage the heterogeneity of data formats and modalities, requiring predefined workflows or extensive manual tuning by domain experts. Despite advances in automating specific components, such as feature engineering [13, 53], neural architecture search [39, 18], and hyperparameter optimization [75], such systems still fall short of providing an end-to-end solution that spans the entire machine learning lifecycle, from data preprocessing to model building. These challenges leave a critical gap for truly autonomous and versatile systems.

The emergence of LLMs brings new opportunities to enhance the flexibility of AutoML. Early attempts to leverage LLMs for AutoML targeted specific pain points, such as LLM-guided feature engineering [29, 82], hyperparameter optimization [47, 79], neural architecture search [35, 55],

39th Conference on Neural Information Processing Systems (NeurIPS 2025).

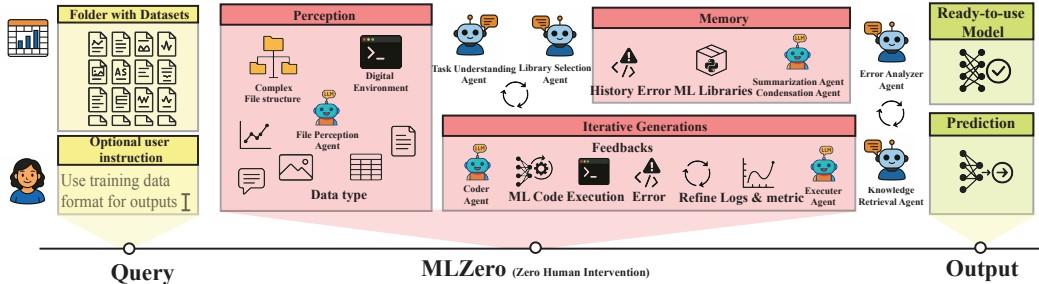

Figure 1: MLZero: An end-to-end multi-agent system that integrates specialized perception agents with dual memory modules (semantic and episodic) to power iterative coding cycles, transforming raw data into ready-to-use models and prediction outputs with zero human intervention.

and data preprocessing [78]. Despite advances, these approaches remain limited by their focus on automating individual components rather than providing truly end-to-end multimodal automation.

With the advent of agentic AI, recent research has explored the broader potential of the LLM agent system for comprehensive AutoML solutions [10, 38, 45, 70, 24], leveraging their demonstrated strengths in planning, code generation [36, 31], and verification processes [81, 37, 73]. However, the success rate of completing complex ML tasks remains hindered by two major challenges: (1) Current systems exhibit a lack of automated data perception, forcing reliance on hard-coded preprocessing logic or rigid data formats [24, 33, 45]. (2) LLMs, when relying solely on their internal parametric knowledge, tend to produce simplistic solutions that fall short for intricate ML tasks [38]. Even with access to advanced ML libraries and domain knowledge, their performance deteriorates when managing long, convoluted knowledge due to the absence of efficient memory mechanisms.

To address these challenges, we introduce MLZero, a novel multi-agent system designed specifically for end-to-end automation of multimodal ML tasks. As illustrated in Figure 1, our approach begins with specialized agents performing comprehensive perception actions to analyze data structures, interpret task requirements, and select the proper ML library, thus establishing a rich perception context for subsequent operations. Building upon the perception context, the system leverages the semantic memory with condensed, curated knowledge about the library to guide iterative code development, while simultaneously utilizing episodic memory for efficient error detection and correction. The dual memory modules enable powerful learning capabilities [83], allowing the system to retain domain knowledge persistently while adaptively learning from past errors, thereby continuously refining its solution through iterations. By seamlessly integrating the reasoning capabilities of LLMs with automated perception and structured memories, MLZero not only overcomes the inherent limitations of previous LLM-based approaches, but also represents a truly end-to-end system that consistently demonstrates superior performance across diverse machine learning tasks. Our main contributions are as follows:

- We introduce MLZero, a novel multi-agent system that delivers high quality end-to-end multimodal machine learning solutions with minimal human intervention. Our system automates the entire workflow through the perception module, dual memory modules (semantic and episodic), and iterative code generation, coordinated across nine specialized agents. Our primary contribution lies in the synergistic integration of these components into a hierarchical multi-agent system specifically designed for end-to-end AutoML on raw, multimodal data.
- MLZero demonstrates superior performance on the existing benchmark, MLE-Bench Lite, with more medal counts (6 gold medal) and higher success rate.
- We benchmark MLZero with a comprehensive benchmark suite, the Multimodal AutoML Agent Benchmark, comprising diverse datasets that incorporate challenging scenarios including multilingual, multitable, multilabel, zero-shot inference tasks with large unprocessed data. Our comprehensive empirical evidence from the constructed multimodal benchmark showing that MLZero outperforms existing ML agents across all evaluated metrics, such as success rate (**+263.6%**), average rank, and time complexity,
- Through detailed ablation studies, we identify key components driving these performance gains and analyze failure cases to demonstrate both the common failures MLZero overcomes and opportunities for future research directions.

## 2 Related Work

### 2.1 LLM Agents for ML

Beyond component-specific applications, several frameworks have emerged for end-to-end LLM-based ML automation [30, 33, 23, 24, 45, 38]. MLAgentBench [33] introduces a research agent that conducts ML experiments by accessing files and executing code. DS-Agent [24] implements case-based reasoning to leverage human expertise, but requires extensive dataset-code pairs during development and struggles with unfamiliar domains. Both approaches lack effective perception capabilities and require code skeleton samples, limiting their end-to-end functionality. AutoKaggle [45] offers a multi-agent system with iterative development and testing, but is restricted to tabular data with hard-coded input formats. AIDE [38] frames ML engineering as a code optimization problem with tree search in the solution space [14, 46] but underperforms without sufficient external knowledge about ML Libraries. Despite these advances, challenges remain in developing truly end-to-end ML automations that operate on raw data with output specifications and minimal human intervention. Key limitations include maintaining consistency across complex workflows, utilizing memory efficiently, adapting to evolving ML libraries, etc. For a more comprehensive evaluation, we deliberately strengthen the existing approaches [38, 24, 45] by prompting them with selected outputs from MLZero (Appendix C).

Meanwhile, MLE-bench [10] offers valuable but methodologically limited agent-human comparisons, constrained by its reliance on preprocessed rather than raw data and outdated human baselines collected under different computational and algorithmic contexts. These limitations motivate our new benchmark's focus on direct agent-to-agent comparisons for end-to-end ML automation.

### 2.2 Tool Use

Recent advances in LLM-based tool-use agents have significantly expanded AI capabilities. Toolformer [63] pioneered an approach where LLMs learn API usage through self-supervision with minimal demonstrations. ToolLLM [60] scaled this concept by developing a comprehensive framework supporting over 16,000 real-world APIs. Chameleon [49] introduced compositional reasoning with a plug-and-play approach for multimodal QA tasks. OctoTools [48] further introduced a training-free agentic framework with standardized tool cards, hierarchical planning, and efficient execution components. Other notable contributions include ToolDoc [32], which demonstrated that API documentation can effectively replace demonstrations, and ToolkenGPT [26] with tool embeddings to avoid fine-tuning. MM-REACT [76] extended these capabilities to visual reasoning tasks, while HuggingGPT [66] demonstrated effective orchestration of multiple models.

While these systems have made significant progress, they mainly focus on tasks with short context lengths. Considering the autoML tasks often require perceiving a large amount of data, our work extends beyond this paradigm by developing a multi-agent approach specifically designed for the more complex and interconnected tasks inherent in ML automations.

### 2.3 Code Generation with LLMs

Large language models have demonstrated increasingly impressive capabilities in code generation and debugging. Models such as AlphaCode [44], Codex [11], and StarCoder [43] have shown competitive programming performance through effective solution generation and self-verification across multiple programming languages. State-of-the-art LLMs [3, 59, 6, 22, 16] have also achieved remarkable results on code benchmarks [12, 8]. Codex CLI [11] enables terminal-based AI coding assistance with secure sandboxed execution. Despite these advances, significant challenges remain in producing code that is semantically aligned with complex user intent, particularly for ML challenges that require specialized library knowledge and intricate data perception and handling. Our work leverages these fundamental code generation capabilities while addressing their limitations through specialized agents and modular designs with ML library integration specifically tailored for ML tasks.

## 3 Methodology

We present MLZero (Figure 2), a multi-agent system $\mathcal{F}$ that automates end-to-end solutions for multimodal ML tasks. The system processes input data $x$ and optional user inputs $U_{\text{opt}}$ to produce

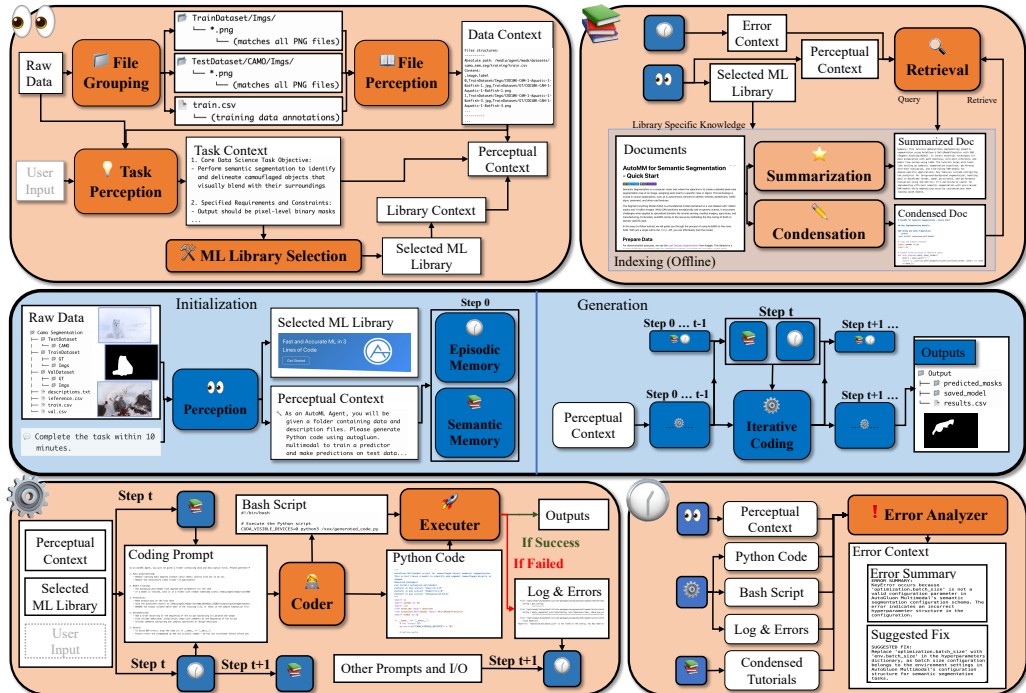

Figure 2: MLZero: A multi-agent system for end-to-end multimodal ML automation with zero human interventions. During the initialization phase, the perception module selects the appropriate ML library and generates perceptual context to initialize semantic and episodic memory. In the subsequent generation phase, the system performs code generation, execution, and debugging iteratively with the assistance of perceptual context, semantic memory, and episodic memory until successful output is achieved. The surrounding panels detail the four key modules: (1) Perception (upper left) in Section 3.1; (2) Semantic Memory (upper right) in Section 3.2; (3) Episodic Memory (lower right) in Section 3.3; and (4) Iterative Coding (lower left) in Section 3.4.

solutions including predicted outputs $y$, code artifacts $C$, and execution logs $L$:

$$\mathcal{F}(x, U^{\text{opt}}) = (y, C, L). \tag{1}$$

Note that our ML model building process for various tasks is achieved by generating code employing different ML libraries and executing it. For supervised learning tasks, $x$ typically includes labeled training data, unlabeled test data, and a brief task description or instruction. For zero-shot tasks, $x$ would simply consist of unlabeled test data and the task description. For example, in a supervised semantic segmentation task as illustrated in Figure 2, $x$ includes training images and the corresponding masks, a tabular file indicating the image-mask pairs, testing images, a tabular file listing the test images, and a simple one-sentence task description. The output $y$ would be the predicted masks for the test images, together with a tabular file that stores the test image-mask pairs. During the evaluation, we calculate the task-specific objective $\mathcal{L}_x$ on prediction $y$ and ground truth $\hat{y}$, that is, $\mathcal{L}_x(y, \hat{y})$.

Our system comprises four modules, where each module is a subsystem with one or more agents, and each agent is a specialized LLM augmented with utility functions (details of each agent in Appendix B): (1) **Perception** that interprets arbitrary data inputs and transforms them into structured context; (2) **Semantic Memory** that enriches the system with knowledge of the ML Library; (3) **Episodic Memory** that maintains chronological execution records for targeted debugging; and (4) **Iterative Coding** that implements a refinement process with feedback loops and augmented memory.

## 3.1 Perception 👀

The Perception module $\mathcal{P}$ acts as the cognitive lens of the system (Eq. 2), orchestrating the transformation of various data inputs into actionable ML workflow specifications. Input is the raw data folder

and optional user input. The output is the perceptual context $P$ and the selected ML library $M$:

$$\mathcal{P}(x, U^{\text{opt}}) = (P, M). \tag{2}$$

This module consists of three agents. **File grouping and file perception agent** (Appendix B.1.1) performs structural analysis of raw data $x$, grouping similar files and interpreting file contents. **Task perception agent** (Appendix B.1.2) extracts semantic information from raw data, derived context, and user input $U_{\text{opt}}$ to identify objectives, constraints, and evaluation criteria, generating task context in natural language. **ML Library selection agent** (Appendix B.1.3) employs context-aware reasoning to match problem characteristics with the appropriate ML Library $M$ by analyzing the constructed task context against the library capabilities. This integrated approach efficiently translates user requirements and dataset properties into perceptual context $P$ that guides subsequent iterative processes.

## 3.2 Semantic Memory 📚

The Semantic Memory Module $\mathcal{S}_t$ enhances the LLM's parametric knowledge with domain-specific information from external knowledge bases at each iteration $t$. These knowledge bases are constructed offline by two agents. The **summarization agent** (Appendix B.2.2) compresses relevant knowledge into concise paragraphs serving as queryable indices, while the **condensation agent** (Appendix B.2.1) transforms this knowledge into precise and streamlined guidance. This agent-based knowledge processing enables the Semantic Memory to incorporate documentation, API references, code examples, and other relevant resources.

At each iteration $t$, given the error context $R_t$ (both Error Summary and Suggested Fix), the Semantic Memory Module processes this information through its **retrieval agent** (Appendix B.2.3) to query the knowledge base of the selected ML library $M$, extracting condensed information $G_t$:

$$\mathcal{S}_t(P, M, R_t) = G_t. \tag{3}$$

In our case, we use tutorial documents to demonstrate the effectiveness of minimal tool initialization. By retrieving contextually relevant tutorials according to the current task and errors encountered, the system provides targeted guidance for iterative coding, substantially improving code quality while reducing LLM hallucinations (Section 4.3.1).

## 3.3 Episodic Memory ⓘ

The Episodic Memory module, $\mathcal{E}_t$, enhances the success rate of MLZero in ML model building by providing error context $R_t$ at each iteration $t$ leveraging its chronological record of the system execution history.

$$\mathcal{E}_t(P, C_{t-1}, L_{t-1}, G_{t-1}, R_{t-1}) = R_t. \tag{4}$$

This component is initialized with the perception context $P$ and progressively stores the interaction data at each iteration. The stored information includes code artifacts $C_{t-1}$, code execution logs $L_{t-1}$, retrieved knowledge $G_{t-1}$, and previous error context $R_{t-1}$. When invoked during code generation, the **error analyzer agent** (Appendix B.3.1) distills encountered issues and contexts into concise error summaries paired with fix suggestions. This focused approach enables subsequent coding agents to efficiently address specific problems without processing excessive contextual information, significantly reducing token consumption while maintaining problem-solving effectiveness (Section 4.3.2).

## 3.4 Iterative Coding ⚙️

With the support of components above, our system enters an iterative coding process $\mathcal{G}_t$, where at each iteration $t$ it refines the solution based on execution feedback:

$$\mathcal{G}_t(P, U_t^{\text{opt}}, R_t, G_t) = (y_t, C_t, L_t). \tag{5}$$

For each iteration $t$, the system first combines the perceptual context $P$, optional user input $U_t^{\text{opt1}}$, error context $R_t$, and the retrieved knowledge $G_t$ to guide the **coder agent** (Appendix B.4.1) in producing executable code $C_t$, leveraging parametric, historical, and external knowledge simultaneously. The

---

[1] Per-iteration user input is disabled in all experiments to show the performance of zero human intervention.

system then executes the generated code in a configured environment, which the coder can further customize or recreate as needed. During execution, the system captures logs $L_t$ including standard output, standard error, log messages, etc., and stores the model output $y_t$. The **executer agent** (Appendix B.4.2) analyzes these results and logs to determine the next steps: finalizing output $y, C, L$ upon success or identifying errors and initiating the next coding iteration $t + 1$.

This iterative approach continues until either successful execution or a maximum iteration limit is reached. Notably, the system supports optional per-iteration user input, allowing for human guidance when required while maintaining a high degree of automation. Through this comprehensive system, MLZero effectively bridges the gap between noisy raw data inputs and sophisticated ML solutions, providing a truly end-to-end automated ML framework adaptive to any modalities.

## 4 Experiments

To evaluate the effectiveness of MLZero against state-of-the-art ML and coding agents, we conducted extensive experiments across multiple benchmarks and datasets. We first assess performance on MLE-bench Lite [10] with 21 diverse Kaggle competitions, then proceeded to the Multimodal AutoML Agent Benchmark for end-to-end evaluation across 25 diverse datasets spanning various modalities and ML tasks. We evaluate the performance using multiple metrics including success rate, average rank, relative time consumption, and solution quality. Additionally, we performed ablation studies to quantify the contribution of individual components within our proposed system. Finally, we conducted a detailed error analysis to identify and categorize failure cases across high-performance methods, providing insights into the robustness and limitations of each approach, followed by efficiency and robustness analysis examining token consumption, cost effectiveness, and robustness across different LLM backbones and under various noise conditions.

### 4.1 Implementation Details

Each agent was assigned a 3-hour time limit per dataset to produce results. By default, MLZero uses Claude 3.7 Sonnet as its underlying LLM. It is important to note that only MLZero and Codex CLI operate truly end-to-end, while other agents require varying degrees of preprocessing or postprocessing to function on our benchmark. For example, DS-Agent requires manual code execution, while results from AIDE and AutoKaggle needed manual extraction from working directories. Complete implementation specifications, including model configurations, the prompts used, and other details for each agent, are provided in the Appendix C.

### 4.2 Main Results

#### 4.2.1 MLEbench

We first evaluate MLZero on MLE-bench Lite [10], consisting of 21 diverse challenges including classification, regression, and generation, spanning various data modalities including image, text, tabular, and audio (more details: Appendix A.2). The results are shown in Figure 3. MLZero demonstrates superior performance on MLEbench-Lite with an average rank of 1.43, significantly outperforming competing approaches including AIDE [38] (2.36), ResearchAgent (MLAB) [33] (3.29), and CodeActAgent (OpenHands) [72] (2.93). Furthermore, MLZero achieves the highest success rate of 86%, compared to AIDE (81%), MLAB (62%), and OpenHands (71%). This performance advantage extends across multiple evaluation metrics, where our approach achieves six gold and two silver medals, while also outperforming competitors in medal counts. Many tasks in MLEBench Lite cannot be handled by the current integrated ML libraries, thus registering more specific ML libraries, e.g. diffusion or audio models, to MLZero can further push the performance.

While MLE-Bench [10] provides valuable comparisons between ML agents and human performance in Kaggle competitions, it relies on dedicated Python scripts to preprocess the raw data, presenting structured rather than raw inputs. This dataset-specific preprocessing significantly simplifies the ML workflow and may overestimate agent capabilities in real-world scenarios. Therefore, to enable more accurate agent-to-agent comparisons in more rigorous end-to-end ML automation, we further tested MLZero on our Multimodal AutoML Agent Benchmark, which specifically challenges agents to handle completely raw, unprocessed data without any task-specific preprocessing assistance.

#### 4.2.2 Multimodal AutoML Agent Benchmark

Table 1: Performance comparison across ML agents and *ablation configurations on Multimodal AutoML Agent Benchmark. Colorful icons indicate data modality ( Tx: text, Tb: tabular, Im: image, Dc: document) and problem type (BC: binary classification, MC: multiclass classification, RG: regression, FC: forecasting, SS: semantic segmentation, RT: retrieval, ML: Multi Label Classification). ↑(↓) indicates higher(lower) is better. †Dagger columns are not agents, including MLZero's optimal performance with current ML libraries to compare with human performance reported in recent open sourced literature. **Bold** numbers indicates best ML agent performance with comparable **def**ault configurations. For each dataset, results are reported as mean$_{std}$ format across three independent runs. Configuration descriptions: **def**: default settings of each agent, **8B**\*: using LLama 3.1 8B [22], **-ext**\*: without external knowledge, **-epi**\*: without episodic memory, **+rea**\*: with reasoning LLM [58], **+ext**\*: with external knowledge. Performance metrics (Appendix A.1.13) are computed across three runs: **Avg. Rank**: average position when ranking agents by their mean valid performance on each dataset, **Rel. Time**: relative execution time compared to MLZero (def), **Success**: percentage of successful runs. Detailed results for each run and each agent are shown in Appendix E.1.

| Agent | End-to-End MLZero (ours) | | | | | End-to-End Codex CLI | | End-to-End AIDE | | DS-Agent | | AK | Human |
|---|---|---|---|---|---|---|---|---|---|---|---|---|---|
| Dataset | def | 8B* | -ext* | -epi* | 24hrs† | def | +rea* | def | +ext* | def | zero-shot | def | Reported† |
| abalone Tb RG↓ | **2.13**$_{.01}$ | 2.09$_{.00}$ | 2.19$_{.02}$ | 2.13$_{.01}$ | 2.08 | 2.23$_{.00}$ | 2.27$_{.06}$ | 2.16$_{.04}$ | 2.18$_{.04}$ | 2.24$_{.00}$ | 2.36$_{.00}$ | × | × |
| electric(H) Tb FC↓ | **1.42**$_{.02}$ | × | 1.75$_{.00}$ | 1.40$_{.01}$ | 1.30 | × | × | × | × | × | 11.66$_{.00}$ | × | 1.23 |
| nn5(D) Tb FC↓ | **0.76**$_{.00}$ | × | 1.14$_{.32}$ | 0.76$_{.00}$ | 0.79 | × | × | × | × | 4.68$_{.00}$ | × | × | 0.76 |
| solar(10m) Tb FC↓ | 1.49$_{.78}$ | × | × | 1.29$_{.00}$ | 0.18 | × | 1.29$_{.00}$ | **1.05**$_{.00}$ | × | × | × | × | × |
| yolanda Tb RG↓ | **8.53**$_{.00}$ | 8.54$_{.00}$ | 8.93$_{.03}$ | 8.53$_{.00}$ | 8.18 | × | 9.43$_{.24}$ | × | 8.79$_{.25}$ | × | × | × | × |
| airbnb Tx Tb MC↑ | **0.43**$_{.00}$ | 0.42$_{.01}$ | 0.24$_{.16}$ | 0.42$_{.01}$ | 0.45 | × | 0.39$_{.00}$ | 0.39$_{.00}$ | 0.39$_{.03}$ | × | 0.31$_{.00}$ | 0.32$_{.05}$ | × |
| airlines Tb BC↑ | **0.66**$_{.00}$ | 0.69$_{.04}$ | 0.63$_{.01}$ | 0.66$_{.00}$ | 0.66 | × | 0.63$_{.02}$ | × | 0.65$_{.01}$ | × | 0.61$_{.00}$ | × | × |
| bio Tb BC↑ | 0.81$_{.01}$ | 0.80$_{.00}$ | 0.83$_{.04}$ | 0.80$_{.00}$ | 0.82 | 0.79$_{.00}$ | 0.85$_{.04}$ | **0.87**$_{.00}$ | 0.84$_{.04}$ | 0.79$_{.00}$ | 0.79$_{.00}$ | × | × |
| camoseg Im SS↑ | **0.84**$_{.00}$ | × | 0.46$_{.00}$ | 0.84$_{.00}$ | 0.85 | × | × | × | × | × | × | × | 0.89 |
| cd18 Im MC↑ | **0.46**$_{.02}$ | -0.21$_{.65}$ | -1.57$_{1.85}$ | 0.51$_{.00}$ | 0.15 | -0.94$_{.53}$ | -1.44$_{.17}$ | × | -0.05$_{.29}$ | -1.94$_{.00}$ | -0.64$_{.98}$ | -1.84$_{.00}$ | × |
| climate Tx RT↑ | **0.48**$_{.00}$ | × | 0.24$_{.01}$ | 0.48$_{.00}$ | 0.48 | × | 0.20$_{.05}$ | × | × | × | × | × | × |
| covertype Tb MC↑ | **0.98**$_{.00}$ | 0.98$_{.00}$ | 0.88$_{.01}$ | 0.98$_{.00}$ | 0.98 | 0.96$_{.00}$ | 0.92$_{.05}$ | 0.88$_{.00}$ | 0.86$_{.09}$ | 0.96$_{.00}$ | 0.95$_{.00}$ | 0.94$_{.00}$ | × |
| flood Im BC↑ | 0.69$_{.00}$ | 0.69$_{.00}$ | 0.60$_{.09}$ | 0.68$_{.00}$ | 0.80 | × | 0.44$_{.00}$ | **0.71**$_{.01}$ | 0.68$_{.02}$ | × | × | 0.58$_{.00}$ | 0.79 |
| fiqa Tx RT↑ | **0.50**$_{.01}$ | × | 0.22$_{.00}$ | 0.46$_{.00}$ | 0.48 | × | 0.20$_{.00}$ | × | × | × | × | × | 0.78 |
| gnad10 Tx MC↑ | 0.86$_{.04}$ | 0.85$_{.00}$ | 0.88$_{.01}$ | 0.83$_{.00}$ | 0.89 | 0.82$_{.03}$ | 0.85$_{.00}$ | **0.90**$_{.01}$ | 0.58$_{.00}$ | × | 0.80$_{.11}$ | 0.11$_{.00}$ | 0.90 |
| ham10000 Im Tb MC↑ | 0.63$_{.11}$ | 0.57$_{.00}$ | 0.67$_{.00}$ | 0.67$_{.11}$ | 0.56 | 0.48$_{.00}$ | 0.47$_{.03}$ | **0.81**$_{.00}$ | 0.81$_{.03}$ | × | × | × | 0.61 |
| hateful Tx Im BC↑ | **0.59**$_{.01}$ | 0.57$_{.04}$ | 0.35$_{.04}$ | 0.59$_{.01}$ | 0.61 | × | 0.48$_{.07}$ | 0.51$_{.05}$ | 0.49$_{.03}$ | × | × | 0.36$_{.02}$ | 0.60 |
| isic2017 Im SS↑ | **0.75**$_{.00}$ | × | × | × | 0.76 | × | 0.11$_{.00}$ | × | × | × | × | × | 0.78 |
| funding Tx BC↑ | **0.45**$_{.02}$ | 0.41$_{.06}$ | 0.36$_{.06}$ | 0.44$_{.00}$ | 0.51 | × | 0.34$_{.04}$ | × | 0.44$_{.00}$ | × | × | 0.24$_{.00}$ | 0.61 |
| memotion Tx Im Tb ML↑ | 0.50$_{.01}$ | × | 0.83$_{.17}$ | 0.51$_{.00}$ | 0.56 | **0.53**$_{.00}$ | 0.76$_{.06}$ | 0.47$_{.00}$ | × | × | × | × | × |
| mldoc Tx MC↑ | 0.95$_{.00}$ | 0.95$_{.00}$ | 0.94$_{.02}$ | 0.95$_{.00}$ | 0.97 | 0.32$_{.23}$ | 0.82$_{.08}$ | **0.96**$_{.00}$ | 0.94$_{.01}$ | 0.95$_{.00}$ | × | × | × |
| petfinder Tx Tb MC↑ | **0.39**$_{.00}$ | 0.40$_{.01}$ | 0.38$_{.02}$ | 0.39$_{.00}$ | 0.41 | × | 0.40$_{.00}$ | 0.34$_{.00}$ | 0.38$_{.01}$ | 0.36$_{.01}$ | 0.27$_{.08}$ | 0.39$_{.00}$ | 0.41 |
| roadseg Im SS↑ | **0.47**$_{.00}$ | × | 0.31$_{.13}$ | 0.60$_{.00}$ | 0.47 | × | × | × | × | × | × | × | 0.62 |
| rvlcdip Dc MC↑ | **0.87**$_{.00}$ | × | 0.89$_{.00}$ | 0.87$_{.00}$ | 0.92 | × | × | × | × | × | × | × | 0.96 |
| clothing Tx Tb RG↑ | **0.75**$_{.00}$ | 0.61$_{.11}$ | 0.66$_{.03}$ | 0.72$_{.04}$ | 0.77 | × | 0.35$_{.23}$ | × | 0.75$_{.00}$ | × | 0.35$_{.00}$ | × | 0.74 |
| **Avg. Rank↓** | 2.42 | 5.14 | 4.94 | 2.86 | N/A | 8.04 | 5.76 | 6.16 | 6.02 | 8.26 | 8.12 | 8.28 | N/A |
| **Rel. Time↓** | 1.0 | 3.17 | 2.32 | 1.03 | N/A | 0.15 | 0.23 | 2.83 | 2.48 | N/A | N/A | 4.82 | N/A |
| **Success↑** | 92.0% | 45.3% | 69.3% | 86.7% | N/A | 14.7% | 69.3% | 25.3% | 45.3% | 13.3% | 20.0% | 14.7% | N/A |

Table 1 provides a detailed performance comparison between MLZero and several state-of-the-art machine learning and coding agents on our Multimodal AutoML Agent Benchmark (further details in Appendices C, and A.1). Evaluated across 25 diverse datasets, MLZero demonstrates a significant advantage. For each dataset, we conduct three independent runs and report the mean performance and standard deviation of valid submissions. We further evaluate the agents based on average rank, relative time consumption, and success rate (see Appendix A.1.13 for definition details).

MLZero with **def**ault configuration achieves a markedly higher success rate (92.0%) compared to all competing agents. Furthermore, MLZero consistently delivers solutions of superior quality, as evidenced by its substantially lower average rank (2.42). In addition to the primary evaluation, we test MLZero with a smaller LLM. The **8B** configuration, which uses LLama 3.1 8B [22], maintains better performance (45.3% success rate, 5.14 rank average) than other agents despite a significantly reduced model size. This demonstrates the robustness of our multi-agent system design. In terms of time complexity, MLZero is significantly faster than AIDE and AutoKaggle due to the more efficient design reducing unnecessary iterations, although slower than Codex CLI, which excels in instruction following and success rate, but tends to give simple solutions that can be executed in a few minutes with degraded performance. Additionally, in the 24-hour extended run-time experiment **24hrs**, we

manually extend the time constraints and improve the quality preset [19, 67, 65] to show that MLZero can approach expert-level performance (Human Reported) given sufficient computational resources.

While Codex CLI shows limited performance with its **def**ault setting using a general-purpose LLM [56], its performance improves substantially with the **+rea** configuration where it is equipped with state-of-the-art reasoning models [58]. Without step limitations and with terminal access, Codex CLI produces simple solutions that execute quickly and iterate efficiently, enabling easier error correction. This approach yields a high success rate and, consequently, a good ranking. However, its performance remains low on challenging datasets, revealing limitations in complex problem-solving.

Considering that each baseline method has different designs regarding the use of external knowledge, we conduct different augmentations to ensure a fair comparison and evaluate methods on equal footing. To assess our method without the advantage of its integrated external knowledge, the **-ext** configuration removes all access to external ML libraries. This configuration yields a 69.3% success rate and a 4.94 average rank, still outperforming all competitors and highlighting the efficiency of MLZero's other components. We also investigate the contribution of episodic memory through the **-epi** configuration, which removes episodic memory but retains the LLM's conversation history within the coder agent. This setup achieves an 86.7% success rate with a 2.86 average rank, demonstrating that while episodic memory provides significant benefits, maintaining a coherent conversational context still yields reasonable performance. In contrast, we explore modifying AIDE to access external knowledge, indicated as **+ext**. While it shows improvement, MLZero continued to outperform it under these comparable conditions. This result underscores that the superior performance of MLZero stems from its overall system design, not merely from the inclusion of episodic memory (**epi**) or external knowledge (**ext**) in isolation.

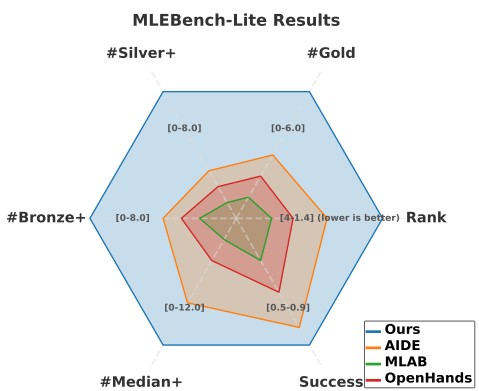

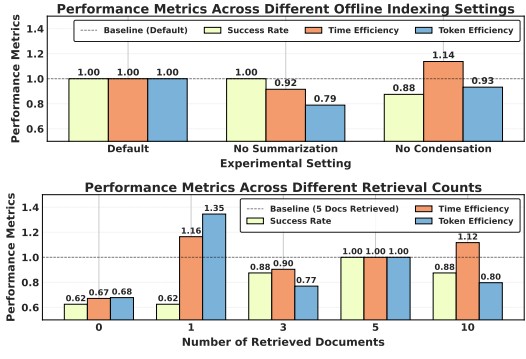

Figure 3: Comparing our agent with baselines on MLE-bench. Detailed results for each run and each agent are shown in Appendix E.2.

Figure 4: Ablation study for semantic memory: Impact on system performance and efficiency of different offline indexing settings (top) and retrieval size (0, 1, 3, 5, 10 documents) (bottom).

## 4.3 More Ablation Studies

To further investigate the effectiveness of an individual component in the overall system, we conduct ablation studies in eight diverse datasets that span tabular data, multimodal data, multilingual and multitable data, document classification, image segmentation, time series forecasting and text retrieval (detailed information available in Appendix A.3). For each ablation experiment, we compare modified configurations with our default system using three primary metrics: (1) success rate in generating valid solutions, (2) token efficiency, and (3) time efficiency. Both token efficiency and time efficiency metrics are calculated as the reciprocal of relative token usage and execution time, respectively, measured only for valid solutions to ensure fair comparisons.

### 4.3.1 More Ablation: Semantic Memory

To further assess our semantic memory module, we evaluated several configurations with varying degrees of functionality. The **def**ault configuration represents our complete system with semantic memory using condensation, summarization, and 5 retrieved documents. **w/o Condensation** uses full tutorials without the condensation process, while **w/o Summary** performs retrieval based only

on tutorial titles, without summary-based indexing. We also evaluated **Retrieval Size** variants by varying the number of documents retrieved (0, 1, 3, 5, 10).

Figure 4 (top) demonstrates that condensation effectively reduces token consumption while preserving performance. Additionally, summary-based retrieval surpasses title-based retrieval, highlighting the importance of content-aware indexing for efficient knowledge access. Figure 4 (bottom) reveals how varying retrieval quantities affects performance. While increasing the number of retrievals generally enhances results, benefits level off after five documents. The token and time usage increase nearly linearly with retrieval quantity, creating a distinct trade-off between performance and computational efficiency. In the zero-retrieval configuration, only the library name is provided to subsequent agents, which then rely solely on the LLM's parametric knowledge of the library.

These findings validate our design choices for the semantic memory module, demonstrating that the combination of condensed tutorials, summary-based retrieval, and an appropriate retrieval size (5 documents) achieves an optimal balance between performance and resource utilization.

Table 2: Ablation study of episodic memory components. Results show token efficiency and success rate across three configurations. The **def**ault configuration with both error summaries and suggested fixes achieves optimal success rate while maintaining reasonable token efficiency.

| Configuration | Token Eff. | Success |
|---|---|---|
| def | 1.0 | **1.0** |
| w/o Fix | 0.81 | 0.75 |
| w/o Sum & Fix | **1.39** | 0.83 |

Table 3: Comparative error analysis of ML agents across different failure categories. Error types include perception, preprocessing, API hallucination, algorithm implementation, and postprocessing phases.

| Error Type | MLZero | Codex CLI +rea | AIDE | DS |
|---|---|---|---|---|
| Perception | 0.0% | 7.7% | 2.1% | 24.0% |
| Preprocessing | 2.1% | 0.0% | 0.0% | 20.0% |
| API Hallucination | 0.0% | 3.8% | 28.2% | 4.0% |
| Algorithm Impl. | 4.3% | 3.8% | 4.3% | 28.0% |
| Postprocessing | 0.0% | 11.5% | 8.7% | 0.0% |
| Unreported | N/A | N/A | 8.7% | N/A |
| Overall | 6.5% | 26.9% | 47.8% | 76.0% |

### 4.3.2 More Ablation: Episodic Memory

To evaluate the contribution of episodic memory, we compared three configurations as shown in Table 2. The **def**ault configuration with both error summaries and suggested fixes achieves the best overall success rate (1.0). The **w/o Sum & Fix** configuration removes both components, prompting the subsequent coder with truncated error messages and executer's analysis instead. It shows token efficiency advantages at the cost of decreased success rate (0.83). Interestingly, the **w/o Fix** configuration, retaining error summaries but removing suggested fixes, has the lowest success rate (0.75) with the worst token efficiency due to the increased number of iterations used to figure out the solution, suggesting that providing over-condensed error summaries without actionable fix suggestions may further confuse the coder. These results indicate that complete episodic memory produces better outcomes, particularly for complex problems requiring multiple debugging cycles.

### 4.4 Failure Cases and Analysis

Among the cases that fail, we conduct an error analysis across MLZero, Codex CLI +rea, AIDE, and DS-Agent (DS), categorizing failures as shown in Table 3. MLZero demonstrates remarkable robustness with minimal errors (6.5% overall in 46 datasets), while competing methods exhibit higher failure rates: Codex CLI (26.9%), AIDE (47.8%), and DSA (76.0%). Notably, MLZero completely eliminates several error categories, including perception, postprocessing, and API Hallucination. It is important to note that error frequencies may be influenced by system progression, i.e. systems that fail earlier (e.g., at perception/preprocessing) never encounter later-stage errors that more capable systems might face when attempting more complex solutions. A detailed discussion of error patterns and their implications for future ML agent design is provided in Appendix E.3.

### 4.5 Efficiency and Robustness Analysis

To assess computational efficiency, we compared token consumption and cost across systems on the benchmark. As shown in Table 4 (left top), MLZero achieves superior performance (92% success

rate) while using 4.3× fewer tokens than AIDE (63k vs. 273k per dataset on average) and incurring only 23% of its cost ($0.19 vs. $0.82 per dataset). The **w/o Fix** configuration demonstrates further token reduction (51k) but at the expense of success rate.

We further evaluated robustness across different LLM backbones (Claude Sonnet 3.7, GPT-4.1, and Llama 3.1 8B) on the eight challenging datasets spanning all modalities (Table 4, left bottom). Results show consistent performance across all models, with the compact 8B model still outperforming all baseline systems from Table 1. Note that X indicates execution timeout or failure, and scores with ↓ indicate lower is better for regression tasks.

To test resilience against noisy inputs, we conducted stress tests with deliberately misleading dataset descriptions on the Abalone tabular regression task (Table 4, right). The table shows different noise conditions: misleading information in the dataset description (Noise Des.), incorrect instructions from the user (Noise Ins.), and removal of the correct library from available options (-Lib). The results columns indicate whether the perception module correctly identified the task (Perc.) and whether the overall end-to-end task was completed successfully (Succ.). Even when provided incorrect task type hints or library suggestions, the perception module correctly identified task characteristics in all cases. When explicitly directed to use incorrect tools via user instruction, the system successfully recovered through restart mechanisms, demonstrating robust error detection and correction capabilities.

Table 4: Efficiency, robustness, and stress test results. Left top: token efficiency and cost. Left bottom: robustness across LLM backbones. Right: stress tests with misleading inputs.

| System | Succ. | Tokens | Cost |
|---|---|---|---|
| MLZero (def) | 92% | 63k | $0.19 |
| MLZero (w/o Fix) | 75% | 51k | $0.15 |
| AIDE (+ext) | 45.3% | 273k | $0.82 |

| Dataset | Sonnet 3.7 | GPT-4.1 | Llama 8B |
|---|---|---|---|
| camoseg | 0.84 | 0.82 | X |
| flood | 0.69 | 0.64 | 0.69 |
| fiqa | 0.50 | 0.50 | 0.50 |
| mldoc | 0.95 | 0.96 | 0.95 |
| petfinder | 0.39 | 0.40 | 0.40 |
| rvlcdip | 0.87 | 0.87 | X |
| solar (10m)↓ | 1.49 | X | X |
| yolanda↓ | 8.53 | 8.53 | 8.54 |

| Noise Des. | Noise Ins. | -Lib | Perc. | Succ. |
|---|---|---|---|---|
| "This is a timeseries task" | × | × | ✓ | ✓ |
| "YOU SHOULD treat this as a timeseries task" | × | × | ✓ | ✓ |
| "IMPORTANT: Use a TimeSeries ML Library" | × | × | ✓ | ✓ |
| "Try to use FlagEmbedding" | × | × | ✓ | ✓ |
| "YOU SHOULD treat this as a timeseries task" | × | ✓ | ✓ | ✓ |
| "Try to use FlagEmbedding" | × | ✓ | ✓ | ✓ |
| None | ✓ | ✓ | × | ✓ |

## 5  Conclusion

In this paper, we introduced MLZero, a novel hierarchical multi-agent system that reimagines multimodal ML automation through the integration of multiple LLM agents with perception, semantic and episodic memory, and iterative coding mechanisms. Our comprehensive evaluation on MLE-bench Lite demonstrates superior performance across 21 diverse Kaggle competitions. Further rigorous end-to-end evaluation on our Multimodal AutoML Agent Benchmark across 25 diverse datasets confirms that MLZero significantly outperforms existing approaches with superior solution quality and a +263% improvement in success rate, while our perception module enables truly end-to-end automation without human intervention. This addresses the limitations of existing benchmarks and provides a more accurate assessment of agent capabilities on raw, unprocessed data.

**Broader Impact**: We believe that this advancement facilitates the democratization of sophisticated ML methodologies, enabling individuals with limited domain expertise to address complex ML challenges effectively. The environmental impact of running LLMs warrants consideration, which presents the need for training smaller LLMs to achieve comparable performance. More limitations and future work can be found in Appendix G.

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

# Contents

# A   Dataset Details

## A.1   Details of Multimodal AutoML Agent Benchmark

Table 5: Dataset information with modality, problem types, evaluation metrics, and data splits

| Dataset | Modality | | | | Problem Type | | | | | | | Metric | Higher is better | Data Samples | | |
|---|---|---|---|---|---|---|---|---|---|---|---|---|---|---|---|---|
| | Tabular | Image | Text | Document | Regression | Multiclass | Binary | Semantic Seg. | Forecasting | Retrieval | Multi-label | | | Train | Val | Test |
| abalone | ✓ | | | | ✓ | | | | | | | RMSE | ✗ | 3,759 | - | 418 |
| airbnb | ✓ | | ✓ | | | ✓ | | | | | | F1(weighted) | ✓ | 18,316 | - | 4,579 |
| airlines | ✓ | | | | | | ✓ | | | | | AUROC | ✓ | 485,444 | - | 53,939 |
| bio | ✓ | | | | | | ✓ | | | | | AUROC | ✓ | 3,375 | - | 376 |
| camoseg | | ✓ | | | | | | ✓ | | | | $S_\alpha$ | ✓ | 3,636 | 404 | 250 |
| cd18 | ✓ | ✓ | | | ✓ | | | | | | | $R^2$ | ✓ | 2,533 | - | 632 |
| climate | | | ✓ | | | | | | | ✓ | | Recall@10 | ✓ | - | - | 1,535 |
| covertype | ✓ | | | | | ✓ | | | | | | F1(weighted) | ✓ | 522,910 | - | 58,102 |
| electric(H) | ✓ | | | | | | | | ✓ | | | MASE | ✗ | - | - | 321 |
| flood | | ✓ | | | | | ✓ | | | | | F1 | ✓ | 3,153 | - | 557 |
| fiqa | | | ✓ | | | | | | | ✓ | | Recall@10 | ✓ | 14,166 | 1,238 | 648 |
| gnad10 | | | ✓ | | | ✓ | | | | | | F1(weighted) | ✓ | 8,228 | 1,017 | 1,028 |
| ham10000 | ✓ | ✓ | | | | ✓ | | | | | | F1(weighted) | ✓ | 10,015 | - | 1,512 |
| hateful | | ✓ | ✓ | | | | ✓ | | | | | F1 | ✓ | 7,134 | - | 1,794 |
| isic2017 | | ✓ | | | | | | ✓ | | | | IoU | ✓ | 2,000 | - | 600 |
| funding | ✓ | | ✓ | | | | ✓ | | | | | F1 | ✓ | 89,879 | - | 22,451 |
| memotion | ✓ | ✓ | ✓ | | | | | | | | ✓ | Average Acc | ✓ | 5,593 | - | 1,399 |
| mldoc | | | ✓ | | | ✓ | | | | | | F1(weighted) | ✓ | * | 5,000 | 4,000 |
| nn5(D) | ✓ | | | | | | | | ✓ | | | MASE | ✗ | - | - | 111 |
| petfinder | ✓ | ✓ | ✓ | | | ✓ | | | | | | F1(weighted) | ✓ | 11,994 | - | 2,999 |
| roadseg | | ✓ | | | | | | ✓ | | | | IoU | ✓ | 1,107 | 13 | 48 |
| rvlcdip | | | | ✓ | | ✓ | | | | | | F1(weighted) | ✓ | 320,000 | 40,000 | 40,000 |
| solar(10m) | ✓ | | | | | | | | ✓ | | | MASE | ✗ | - | - | 137 |
| clothing | ✓ | | ✓ | | ✓ | | | | | | | $R^2$ | ✓ | 18,789 | 2,349 | 2,348 |
| yolanda | ✓ | | | | ✓ | | | | | | | RMSE | ✗ | 360,000 | - | 40,000 |

The mldoc dataset has varying training set sizes (1K, 2K, 5K, 10K) across multiple languages (German, English, Spanish, French, Italian).

### A.1.1   Construction

We constructed the Multimodal AutoML Agent Benchmark (MAAB) to address a critical gap in existing benchmarks: the ability to evaluate agents on raw, unprocessed multimodal data. To ensure

fairness and diversity, all 25 datasets are sourced from reputable public repositories including Kaggle competitions, UCI Machine Learning Repository, and the BEIR benchmark suite. Datasets were selected from published machine learning research, with most lacking direct open-source solutions to prevent contamination. The benchmark spans 7 distinct problem types (binary classification, multiclass classification, multi-label classification, regression, segmentation, retrieval, and time series forecasting) across multiple data modalities including tabular, text, image, document, and multimodal combinations. We open-source the complete benchmark including all datasets, evaluation scripts, and preprocessing specifications. Our evaluation follows standard practices: the knowledge base contains only general library documentation with no information about evaluation datasets or their solutions, and test sets remain completely held out during all execution stages.

### A.1.2 Preprocessing

To maintain the realistic challenge of processing raw data while ensuring fair comparison across methods, we implemented minimal preprocessing interventions. Specifically, we: (1) augmented datasets lacking descriptions with brief task specifications to accommodate baseline agent requirements, (2) removed target columns from test files to prevent data leakage to agent.

### A.1.3 Tabular Classification and Regression

We select five diverse datasets (with details below) for tabular classification and regression tasks from TabRepo [61]. To accommodate baseline agent requirements, we augmented each dataset with brief task descriptions. These descriptions provide minimal but sufficient context for evaluation across all comparative systems while ensuring consistency in task. For instance, the abalone dataset includes the description:

> **Description for Abalone (Tabular Classification and Regression)**
>
> ```
> Regression on Class_number_of_rings. Eval metric is RMSE.
> ```

**The Abalone Dataset (abalone)** [54, 61] consists of $3,759$ training instances and $418$ testing instances, totaling $4,177$ samples. Each observation contains 8 input features: sex (categorical: Male, Female, or Infant), and 7 physical measurements including length, diameter, height, whole weight, shucked weight, viscera weight, and shell weight. The regression task involves predicting the number of rings in the abalone shell, which serves as a proxy for the abalone's age. The evaluation metric for this regression task is Root Mean Square Error (RMSE). The dataset presents a challenging regression problem due to the non-linear relationship between the physical measurements and the target variable, making it a valuable benchmark for assessing regression algorithms.

**The Airlines Dataset (airlines)** [61] consists of 485,444 training instances and 53,939 test instances for flight delay prediction. Each instance contains 8 features: airline carrier code, flight number, origin airport, destination airport, day of week (1-7), time of departure (in minutes), flight duration (in minutes), and the binary target variable indicating delay status (0 for on-time, 1 for delayed). The dataset captures commercial flight information across various U.S. airports, carriers, and time periods. This binary classification problem represents a practical application in transportation logistics, where accurate delay predictions can significantly impact resource allocation, scheduling decisions, and customer satisfaction in the aviation industry.

**The Bioresponse Dataset (bio)** [61] contains 3,751 samples (3,375 training, 376 testing) for binary classification of molecular biological responses. Each row represents a molecule with a binary target indicating response presence (1) or absence (0). The feature space consists of 1,776 normalized molecular descriptors (D1-D1776) representing chemical properties such as size, shape, and elemental constitution.

**The Covertype Dataset (covertype)** [9, 61] consists of $522,910$ training samples and $58,102$ testing samples, totaling $581,012$ instances. It represents a multiclass classification problem aimed at predicting forest cover types from cartographic variables without remotely sensed data. The dataset encompasses four wilderness areas within the Roosevelt National Forest in northern Colorado, characterized by minimal human disturbance, thus reflecting primarily ecological processes rather than forest management influences. The target variable classifies areas into seven cover types, including Spruce/Fir, Lodgepole Pine, Ponderosa Pine, Cottonwood/Willow, Aspen, Douglas-fir, and

Krummholz. Predictor variables include elevation, aspect, slope, and various soil characteristics represented as binary features. This dataset is particularly valuable for evaluating algorithmic performance on high-dimensional, imbalanced, multiclass problems with geographic dependencies.

**The Yolanda Dataset (yolanda)** [25, 61] consists of $400,000$ total samples, divided into $360,000$ training instances and $40,000$ testing instances. This dataset was utilized for a regression task, where the objective is to predict the values in column "101" based on the preceding features.

### A.1.4 Multimodal Classification and Regression

We evaluated our system on nine diverse multimodal classification and regression datasets) [67]: airbnb (airbnb melbourne) [1], cd18 (cd18 cellphone) [77], flood (european flood depth) [34], gnad10 [62], ham10000 [71], hateful (hateful meme) [41], funding (kick start funding), petfinder [2], and clothing (women clothing review) [4]. These datasets incorporate various combinations of tabular, text, and image data, presenting distinct challenges for automated machine learning systems. Similarly, we supplemented datasets lacking descriptions with concise task specifications to ensure consistent evaluation across all compared methods.

> **Description for CD18 (Multimodal Classification and Regression)**
>
> ```
> The goal of this research is to achieve an arrangement
> to predict the price of a cellphone based on its specifications.
> ```

**The Airbnb Melbourne Dataset (airbnb)** [1, 24, 67] consists of 22,895 listings, partitioned into 18,316 training samples and 4,579 testing samples. The dataset captures various features of Airbnb accommodations in Melbourne, Australia. The target variable is `price_label`, which represents categorized price ranges grouped into distinct bins. To prevent data leakage, the raw `price` feature has been removed from the dataset. This classification task requires models to predict the price category of a listing based on its characteristics, such as location, property type, amenities, and host attributes, making it suitable for evaluating price prediction algorithms in the short-term housing market.

**Cellphone Dataset (cd18)** [77, 67] contains 2,533 training samples and 632 testing samples, encompassing diverse attributes such as model information, physical characteristics (weight), technical specifications (processor, RAM, storage, display dimensions, resolution), battery information, camera capabilities, operating system, and market metrics (hit count). The dataset spans multiple manufacturers, device generations, and price points, providing the information for developing and evaluating predictive models for cellphone pricing based on technical specifications.

**European Flood Depth Dataset (flood)**     [34, 67] comprises 3,710 images for binary classification, with 3,153 samples in the training set and 557 samples in the test set. Each image is labeled as either "Not useful for determining depth" (0) or "Useful for determining depth" (1), representing the image's utility in flood depth estimation. The dataset includes metadata such as image identifiers, confidence scores, and bounding box coordinates. This dataset serves as a foundation for developing models that can automatically identify imagery suitable for flood depth analysis, potentially enhancing rapid damage assessment capabilities during flooding events.

**The 10kGNAD Dataset (gnad10)** [62, 67] consists of $10,273$ German news articles collected from an Austrian online newspaper, categorized into 9 distinct classes: Web, Panorama, International, Wirtschaft, Sport, Inland, Etat, Wissenschaft, and Kultur. The dataset is split into $8,228$ training samples, $1,017$ validation samples, and $1,028$ testing samples. Article titles and text are concatenated, with author information deliberately removed to prevent classification based on author-specific keywords. It serves as a benchmark for German topic classification tasks, addressing the unique challenges posed by the German language's higher inflection rate and longer compound words compared to English.

**The HAM10000 Dataset (ham10000)** [71, 67] consists of 10,015 training and 1,512 testing dermatoscopic images collected from diverse populations using various acquisition modalities. We removed the `dx_type` feature to prevent potential data leakage in our experimental setup. The dataset encompasses seven diagnostic categories of pigmented skin lesions: actinic keratoses and Bowen's disease (`akiec`), basal cell carcinoma (`bcc`), benign keratosis-like lesions (`bkl`), dermatofibroma

(df), melanoma (`mel`), melanocytic nevi (`nv`), and vascular lesions (`vasc`). Ground truth diagnoses were established through histopathology (>50% of cases), follow-up examination, expert consensus, or in-vivo confocal microscopy. The dataset includes metadata on patient demographics (age, sex), lesion localization, and unique lesion identifiers that allow tracking multiple images of the same lesion.

**The Hateful Memes Dataset (hateful)** [41, 67] is a multimodal benchmark consisting of $7,134$ training and $1,794$ testing examples designed to evaluate hate detection systems. Each sample contains an image paired with overlaid text, requiring models to understand both visual and linguistic content and their potentially harmful interactions. The dataset is labeled binary, where $1$ indicates hateful content (targeting protected categories through harmful stereotypes, derogatory language, or harmful contexts) and $0$ represents benign content. The challenging nature of this dataset stems from the necessity for models to comprehend cross-modal relationships, as the hateful nature often emerges from the interaction between text and imagery rather than from either modality independently.

**The Kickstarter Funding Dataset (funding)** [67] consists of $112,330$ crowdfunding campaign records ($89,879$ for training and $22,451$ for testing) extracted from the Kickstarter platform. Each record contains 10 features including campaign name, description, funding goal amount, keyword identifiers, communication preferences, country of origin, currency type, deadline timestamps, creation timestamps, and the target variable (final_status) indicating campaign success or failure. The binary classification task involves predicting whether a crowdfunding campaign will successfully reach its funding goal based on the provided features. This dataset captures the multifaceted nature of crowdfunding dynamics across various project categories, geographical regions, and funding scales.

**The PetFinder.my Adoption Prediction Dataset (petfinder)** [2, 67] consists of $14,993$ samples ($11,994$ training, $2,999$ testing) designed for predicting pet adoption speed. Each entry contains pet characteristics including type, age, breed, gender, color, size, health status, and vaccination records. Additional features encompass textual descriptions, rescuer information, quantity of photos and videos, adoption fees, and geographical location. The target variable, AdoptionSpeed, is categorized on a scale indicating how quickly pets were adopted. The dataset also includes image paths for visual analysis, making it suitable for multimodal machine learning approaches to predict adoption outcomes.

**The Women's E-Commerce Clothing Reviews Dataset (clothing)** [4, 67] comprises $23,486$ customer reviews of women's clothing products from an e-commerce retailer. Each observation contains 10 features: a unique clothing identifier, reviewer's age, review title, review text, product rating (1-5 scale), binary recommendation indicator, positive feedback count (number of customers finding the review helpful), and three hierarchical product categorization variables (division, department, and class names). For our experiments, we employed an 8-1-1 train-validation-test split of the data. The dataset enables analysis of customer sentiment, recommendation patterns, and demographic preferences across different clothing categories, making it suitable for natural language processing, sentiment analysis, and recommendation system research.

### A.1.5  Document Classification

Document classification entails the categorization of visual documents, which can be approached as an image classification task, an OCR-based text classification task, or as a multimodal problem integrating both methodologies. Our experiments utilize the RVL-CDIP (Ryerson Vision Lab Complex Document Information Processing) [27] dataset, comprising 400,000 grayscale images distributed across 16 distinct document classes with 25,000 images per class. The dataset is partitioned into 320,000 training images, 40,000 validation images, and 40,000 test images. This dataset presents particular challenges due to its substantial size and intricate file structure, highlighting the necessity of effective perception rather than naïve loading of all file structures into the LLM, which would exceed context limitations.

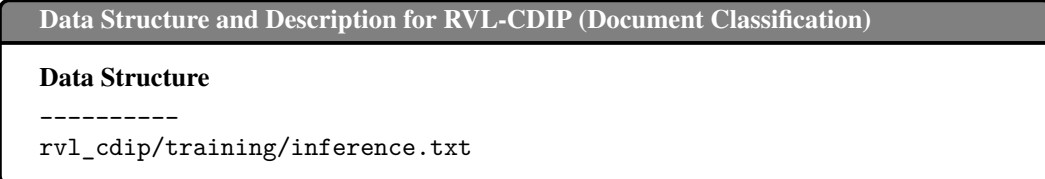

```
Content:
File Size: 1.55 MB
First few lines of the file:
document
imagesr/r/g/e/rge31d00/503210033+-0034.jpg
imagesc/c/e/j/cej80d00/517306722+-6724.jpg
imagesm/m/r/r/mrr36d00/50603620-3621.jpg
----------
rvl_cdip/training/val.txt
Content:
File Size: 1.64 MB
First few lines (up to 1024 characters):
document label
imagesg/g/t/h/gth35e00/2024525661.jpg 11
imagesi/i/y/k/iyk38c00/512015827+-5827.jpg 0
imagesr/r/r/e/rre21e00/87103403.jpg 0
----------
rvl_cdip/training/train.txt
Content:
File Size: 13.09 MB
First few lines:
document label
imagesq/q/o/c/qoc54c00/80035521.jpg 15
imagese/e/w/c/ewc23d00/513280028.jpg 1
imagesw/w/b/t/wbt26e00/2053453161.jpg 7
----------
rvl_cdip/training/readme.txt
----------
Group pattern: rvl_cdip/training/images/*/*/*/*/*/*.tiff
(total 400000 files)
Example file:
rvl_cdip/training/images/imagesu/u/j/j/ujj43a00/87928159.tiff
Content:
File Size: 0.03 MB
Image Format: TIFF
Image Mode: L
Image Size: (754, 1000)
Array Shape: (1000, 754)
Data Type: uint8
Min Pixel Value: 0
Max Pixel Value: 255
Mean Pixel Value: 76.94
Standard Deviation: 116.63
----------
```

**Description (readme.txt)**

```
----------------

RVL-CDIP Dataset

----------------

The RVL-CDIP (Ryerson Vision Lab Complex Document Information
Processing) dataset consists of 400,000 grayscale images in
16 classes, with 25,000 images per class. There are 320,000
training images, 40,000 validation images, and 40,000 test
images. The images are sized so their largest dimension does
not exceed 1000 pixels.
```

```
-------

DETAILS

-------

The label files list the images and their categories in the
following format:

path/to/the/image.tif category

where the categories are numbered 0 to 15, in the following order:

0 letter
1 form
2 email
3 handwritten
4 advertisement
5 scientific report
6 scientific publication
7 specification
8 file folder
9 news article
10 budget
11 invoice
12 presentation
13 questionnaire
14 resume
15 memo

Prediction should be saved in a dataframe with column names "document"
and "label".
```

**The RVL-CDIP Dataset (rvlcdip)** [27] comprises $400,000$ grayscale document images scanned at approximately 100 dpi, distributed across 16 distinct semantic categories including letters, forms, emails, resumes, and memos. The dataset is partitioned into $320,000$ training images, $40,000$ validation images, and $40,000$ test images, with each class being represented equally. The corpus is characterized by significant quality variations typical of real-world document collections, including scanning artifacts, noise, skew, and low resolution, making it a challenging benchmark for document classification tasks.

### A.1.6 Multilingual and Multitable Classification

We employed the mldoc (MLDoc-11000) [64] dataset for evaluating multilingual classification capabilities, extending beyond the monolingual English usage in previous work [67]. Our configuration incorporates multiple languages (German, English, Spanish, French, and Italian) simultaneously. Specifically, we concatenated test files from all five languages to form a multilingual test set while preserving the original structure of the remaining data files.

```
Description for MLDoc-11000 (Multilingual and Multitable Classification)

This is a multi lingual task.
Train a multilingual model and give the predictions.
```

**The Multilingual Document Classification Corpus (mldoc)** [64] is a cross-lingual text classification benchmark. It encompasses eight languages: English, German, French, Spanish, Italian, Russian, Japanese, and Chinese. MLDoc features balanced class distribution across all languages and follows a standardized train/dev/test split. For most languages, training sets of varying sizes (1K, 2K, 5K, and 10K documents) are provided, with Spanish and Russian having 9,458 and 5,216 training documents respectively. Each language includes a 1K development set and a 4K test set. In our experiments, we

utilize the German, English, Spanish, French, and Italian subsets, treating this as both a multi-table and multilingual classification task, with separate training tables for each language and combined test files.

### A.1.7 Multimodal Classification (Multilabel)

We adopted the memotion [50] dataset for multilabel classification. We incorporates all five classification dimensions: humor (Not humorous, funny, very funny, hilarious), sarcasm (Not Sarcastic, general, twisted meaning, very twisted), offensive (Not offensive, slight, very offensive, hateful offensive), motivational (Not Motivational, Motivational), and overall sentiment (Negative, Very Negative, Positive, Very Positive, Neutral). We employ average accuracy as the evaluation metric to assess performance across all dimensions.

---

**Description for Memotion (Multimodal Multilabel Classification)**

```
Train the model on image and corrected text to predict "humour",
"sarcasm", "offensive", "motivational", "overall_sentiment" of the meme.

humour: Not humorous, funny, very funny, hilarious
sarcasm: Not Sarcastic, general, twisted meaning, very twisted
offensive: Not offensive, slight, very offensive, hateful offensive
motivational: Not Motivational, Motivational
overall_sentiment: Negative, Very Negative,
    Positive, Very Positive, Neutral
```

---

**The Memotion Dataset (memotion)**    [50, 67] The Memotion dataset comprises $5,593$ training and $1,399$ testing samples, each consisting of meme images paired with corrected text annotations. This dataset is designed for multi-label classification across five dimensions: humor (four classes: not humorous, funny, very funny, hilarious), sarcasm (four classes: not sarcastic, general, twisted meaning, very twisted), offensiveness (four classes: not offensive, slight, very offensive, hateful offensive), motivation (binary: not motivational, motivational), and overall sentiment (five classes: negative, very negative, positive, very positive, neutral). The dataset enables research on automated understanding of emotional and contextual elements in internet memes, addressing the complex interplay between visual and textual components in multimodal sentiment analysis.

### A.1.8 Semantic Segmentation

We include three semantic segmentation datasets: camoseg (camouflaged segmentation) [42], roadseg (road segmentation) [51], and isic2017 [15]. Each dataset presents unique visual analysis challenges requiring pixel-level prediction capabilities. To standardize evaluation, we provided descriptive task specifications; for instance, the camoseg dataset includes the description:

---

**Description for Camouflaged Segmentation (Semantic Segmentation)**

```
Perform semantic segmentation to identify and delineate
camouflaged objects that visually blend with their surroundings,
outputting pixel-level binary masks.
```

---

**The Camouflage Segmentation (camoseg)** [67, 42, 84] dataset consists of $4,290$ images with corresponding pixel-level binary masks, designed for identifying and segmenting camouflaged objects that visually blend with their surroundings. The dataset is partitioned into training ($3,636$ images), validation ($404$ images), and testing ($250$ images) subsets. Each sample comprises an input image paired with its ground truth segmentation mask, where camouflaged objects are delineated at the pixel level. The dataset encompasses a diverse range of naturally camouflaged subjects, demonstrating various camouflage mechanisms observed in natural environments.

**The ISIC 2017 Dataset (isic2017)** [15, 67, 84] consists of dermoscopic images used for automated melanoma diagnosis, specifically focused on semantic segmentation of skin lesions. The dataset comprises $2,000$ training images and $600$ testing images, each paired with expert-annotated ground truth

segmentation masks, aimed at advancing computer vision techniques for dermatological applications and improving early detection of melanoma and other skin cancers.

**The Massachusetts Roads Dataset (roadseg)** [51, 67, 84] consists of satellite imagery for road segmentation tasks, comprising $1,107$ training samples, 13 validation samples, and 48 testing samples. Each sample consists of an input satellite image paired with its corresponding ground truth road segmentation mask. The dataset is structured to facilitate the development and evaluation of machine learning models for automated road network extraction from aerial imagery. The input images are stored as PNG files in designated directories, with corresponding output masks containing pixel-wise road annotations.

### A.1.9  Retrieval

We incorporated two retrieval datasets from the BeIR benchmark [68]: fiqa (FiQA-2018) and climate (Climate Fever). The Climate Fever dataset serves as a zero-shot evaluation as it lacks a training set, testing systems' ability to perform retrieval without task-specific training data. To optimize computational resources during benchmarking, we reduced the Climate Fever corpus size to one-tenth of its original scale while maintaining the positive documents. In the description file we specify the output format for easier evaluation:

---

**Description for FiQA-2018 (Retrieval)**

```
Finetune the model on training data.
And then retrieve the top 10 results for the queries.
The output should have three columns: query-id, corpus-id, score.
Each row is a retrieved pair.

e.g.
query-id     corpus-id     score
8            566392        0.6
8            65404         0.4
15           325273        0.7
```

---

**The Climate-FEVER Dataset (climate)** [17, 68] is a zero-shot fact verification dataset consisting of $1,535$ testing query-document pairs without training data. The original corpus contains $5,418,128$ documents; however, for computational efficiency, we utilize a 10% subset of the corpus while ensuring all ground truth documents are preserved. The dataset follows the FEVER [69] framework but applies it to real-world climate change-related claims collected from the Internet, offering unique challenges in natural language understanding. For our experiments, we require retrieval output using the standard format of query-id, corpus-id, and relevance score, evaluating the recall for top-10 retrieved documents for each query.

**The FiQA-2018 Dataset (fiqa)** in the BEIR benchmark [68] consists of financial question-answering data with $14,166$ training samples, $1,238$ validation samples, and 648 test samples. The corresponding document corpus contains $57,638$ documents. We evaluated performance by retrieving the top-10 most relevant documents for each query in the test set and compute the recall.

### A.1.10  Timeseries Forecasting

We incorporated three timeseries forecasting datasets [65, 5] with varying temporal resolutions: nn5(D) (nn5_daily), solar(10m) (solar_10_minutes), and electric(H) (electricity_hourly). The training data is provided in compressed JSONL format with json.gz extension. To mitigate potential confusion regarding data format, we supplemented each dataset with detailed format specifications. For example, the nn5_daily dataset includes the following description:

---

**Description for nn5_daily (Timeseries Forecasting)**

```
File Structure:
jsonl, each line is formatted as below:
{   "target":[18.13,25.46,25.11,...],
```

---

```
    "start":"1996-03-18 00:00:00",
    "item_id":"T1",
    "feat_static_cat":[0]
}
You should train models that forecast the target up to 56 days into the
future. Evaluation metric is MASE.
Note that the frequency is 1 day.
The output should be a jsonl file.
```

**The Electricity Hourly Dataset (electric(H))** consists of 321 distinct time series representing electricity consumption measurements collected at hourly frequency. Each series contains $8,428,176$ time steps, spanning multiple years of observations. The forecasting task involves predicting 24 steps (hours) into the future. The dataset is structured in JSONL format, where each record contains a target vector of consumption values, a timestamp indicating the starting point, a unique identifier for each series, and categorical features. Models trained on this dataset are evaluated using the Mean Absolute Scaled Error (MASE) metric.

**The NN5 Daily Dataset (nn5(D))** [65, 5] comprises 111 time series from the banking domain, specifically recording daily cash withdrawal amounts from automated teller machines (ATMs) across the United Kingdom. The dataset was designed to evaluate forecasting methods' efficacy in predicting daily ATM withdrawal patterns. To address data quality concerns, missing values in the original dataset were imputed using a day-of-week median replacement strategy, whereby each missing observation was substituted with the median value calculated from the same weekday across the entire corresponding time series. The forecasting task involves predicting 56 steps (days) into the future.

**The Solar Dataset (solar(10m))** [65, 5] comprises 137 time series representing 10-minute observations of solar power production recorded per every 10 minutes in Alabama state in 2006. The forecasting task involves predicting 60 steps (10-minutes) into the future.

### A.1.11 Evaluation

We implemented a rigorous evaluation protocol to ensure fair and consistent assessment across all systems. As previously noted, we removed target columns from test files to prevent data leakage to agents. The complete test data remained accessible only within the evaluation environment, where we performed strict line-by-line ordered comparisons between agent outputs and ground truth. This protocol enforces format compliance, i.e. deviations in output structure, naming conventions, or completeness result in evaluation failure. To facilitate consistent evaluation while maintaining the challenge of raw data processing, we maintained a metadata file for evaluator reference (not exposed to agents):

**metadata.json for Abalone dataset**

```
{
  "dataset_name": "abalone",
  "metric_name": "rmse",
  "problem_type": "regression",
  "label_column": "Class_number_of_rings",
  "modality": ["tabular"]
}
```

This structured evaluation framework ensures that success metrics reflect not only predictive performance but also practical usability in real-world scenarios where output format requirements are equally critical.

### A.1.12 Human Reported Results

The human reported baseline results are retrieved from established open-source benchmarks and literature: camoseg [67], electric(H) [65], flood [67], fiqa [52], gnad10 [67], ham10000 [67], hateful [67], isic2017 [67], funding [67], nn5(D) [65], petfinder [67], roadseg [67], rvlcdip [7], and clothing [67].

For the remaining datasets, comparable metrics from open-source implementations are unavailable, either due to the absence of published performance metrics, differences in evaluation metrics, or because our modifications to these datasets preclude direct comparison.

### A.1.13 Average Rank, Relative Time, and Success

**The average relative time usage (Rel. Time)** quantifies computational efficiency across models by normalizing execution times against the reference model MLZero (def). For each dataset $d$ in the collection $\mathcal{D}$, we compute the ratio of execution time $t_{m,d}$ for model $m$ relative to the reference model's time $t_{ref,d}$. The average relative time usage $\overline{T}_m$ for model $m$ is then defined as:

$$\overline{T}_m = \frac{1}{|\mathcal{D}_{m,ref}|} \sum_{d \in \mathcal{D}_{m,ref}} \frac{t_{m,d}}{t_{ref,d}}$$

where $|\mathcal{D}_{m,ref}|$ represents the number of datasets with valid measurements for both the evaluated model and the reference model. Both $t_{ref,d}$ and $t_{m,d}$ are averaged across valid results in three runs for each model-dataset combination.

**The average rank (Avg. Rank)** measures the relative performance of each model across all datasets. For each dataset $d$, models are ranked in descending order of performance, with the best-performing model receiving rank 1. Invalid results are assigned the worst rank. The average rank $\overline{R}_m$ for model $m$ is computed as:

$$\overline{R}_m = \frac{1}{|\mathcal{D}|} \sum_{d \in \mathcal{D}} r_{m,d}$$

where $r_{m,d}$ is the rank of model $m$ on dataset $d$, and $|\mathcal{D}|$ is the number of datasets. When multiple models have the same performance score on a dataset, the average mode is used, where tied models receive the same rank calculated as the average of the ranks they would have received had they not been tied.

**The success rate (Success)** indicates the percentage of dataset-run combinations where a model successfully completes the task. For model $m$, the success rate $S_m$ is defined as:

$$S_m = \frac{|\mathcal{D}_m|}{|\mathcal{D}| \times 3} \times 100\%$$

where $|\mathcal{D}_m|$ is the number of dataset-run combinations with valid results for model $m$, and $|\mathcal{D}|$ is the total number of datasets in the evaluation. Each dataset is evaluated across 3 independent runs, resulting in $|\mathcal{D}| \times 3$ total evaluation instances.

## A.2 Details of MLE-bench Lite

### A.2.1 Implementation Details

For the MLE-bench Lite [10] evaluation, we modified our hyperparameter configuration from those used in the Multimodal AutoML Agent Benchmark. Specifically, we configured the agent to use best quality preset parameters when available [19, 65, 67] with a 4-hour time limit per iteration to thoroughly evaluate performance. We established a 24-hour overall limit for each dataset. All experiments were conducted on an AWS EC2 p4d.24xlarge instance equipped with 8 NVIDIA A100 (40GB) GPUs and 96 vCPUs.

Since training requires substantial computational resources and time, we utilize the performance metrics reported in the original MLE-Bench publication [10]. Although the specific LLM model is not explicitly stated in these reports, we select the first reported results for each agent to maintain consistency. We excluded the ranzcr-clip-catheter-line-classification dataset due to preprocessing inconsistencies, ensuring fair comparison across all systems.

Many competitions in the benchmark closed several years ago, making the leaderboard rankings potentially outdated compared to current human capabilities. Therefore, beyond medal counts, we compare the relative performance ranking of each agent across all datasets and compute average rankings. Our evaluation framework incorporates six comprehensive metrics: (1) number of gold medals, (2) number of gold and silver medals combined, (3) number of gold, silver, and bronze medals combined, (4) number of performances above median threshold, (5) success rate (proportion of non-trivial submissions with non-zero performance), and (6) average rank across all datasets.

### A.2.2 Performance Thresholds for MLE-bench Datasets

Table 6 presents the performance thresholds established for each dataset's medal and above-median categorization.

Table 6: Dataset Indices and MLE-Bench Information

| ID | Dataset Name | Gold | Silver | Bronze | Median |
|---|---|---|---|---|---|
| D1 | Aerial Cactus Identification (↑) | 1.0000 | 1.0000 | 1.0000 | 0.9991 |
| D2 | APTOS 2019 Blindness Detection (↑) | 0.9305 | 0.9197 | 0.9145 | 0.8889 |
| D3 | Denoising Dirty Documents (↓) | 0.0179 | 0.0261 | 0.0452 | 0.0733 |
| D4 | Detecting Insults in Social Commentary (↑) | 0.8332 | 0.8231 | 0.7911 | 0.7784 |
| D5 | Dog Breed Identification (↓) | 0.0005 | 0.0054 | 0.0460 | 0.4721 |
| D6 | Dogs vs Cats Redux (↓) | 0.0388 | 0.0504 | 0.0613 | 0.1222 |
| D7 | Histopathologic Cancer Detection (↑) | 0.9835 | 0.9798 | 0.9738 | 0.9477 |
| D8 | Jigsaw Toxic Comment Classification (↑) | 0.9874 | 0.9867 | 0.9864 | 0.9808 |
| D9 | Leaf Classification (↓) | 0.0000 | 0.0079 | 0.0153 | 0.1083 |
| D10 | MLSP 2013 Birds (↑) | 0.9353 | 0.9004 | 0.8737 | 0.8657 |
| D11 | NYC Taxi Fare Prediction (↓) | 2.8338 | 2.8819 | 2.9237 | 3.5974 |
| D12 | NOMAD 2018 Conductors (↓) | 0.0559 | 0.0623 | 0.0658 | 0.0699 |
| D13 | Plant Pathology 2020 FGVC7 (↑) | 0.9784 | 0.9747 | 0.9736 | 0.9485 |
| D14 | Random Acts of Pizza (↑) | 0.9791 | 0.7648 | 0.6921 | 0.5996 |
| D15 | SIIM-ISIC Melanoma Classification (↑) | 0.9455 | 0.9401 | 0.9370 | 0.9128 |
| D16 | Spooky Author Identification (↓) | 0.1651 | 0.2700 | 0.2938 | 0.4188 |
| D17 | Tabular Playground Dec 2021 (↑) | 0.9566 | 0.9566 | 0.9566 | 0.9534 |
| D18 | Tabular Playground May 2022 (↑) | 0.9982 | 0.9982 | 0.9982 | 0.9727 |
| D19 | Text Normalization English (↑) | 0.9972 | 0.9914 | 0.9904 | 0.9904 |
| D20 | Text Normalization Russian (↑) | 0.9901 | 0.9823 | 0.9759 | 0.9759 |
| D21 | ICML 2013 Whale Challenge (↑) | 0.9896 | 0.9502 | 0.9052 | 0.8652 |

### A.2.3 Limitations of MLE-bench Evaluation

MLE-bench [10] provides valuable comparisons between ML agents and human performance on Kaggle competitions, but has notable limitations. The benchmark uses preprocessed data through dedicated Python scripts, presenting structured rather than raw inputs, which may overestimate agent capabilities. This contrasts with our Multimodal AutoML Agent Benchmark that specifically tests end-to-end processing of raw data. Additionally, MLE-bench's agent-human comparisons face methodological issues: different data splits, evaluation conditions, and human leaderboards that often reflect earlier computational environments with limited hardware and algorithms. These factors introduce systematic biases that complicate performance comparisons and may not accurately represent contemporary ML automation capabilities among human competitors. While MLE-bench demonstrates our system's effectiveness within established workflows, our benchmark evaluates the end-to-end capabilities that distinguish MLZero from other ML agent approaches.

### A.3 Details of Datasets used in Section 4.3 for Ablation Studies

To evaluate the contribution of individual components to our system's overall performance, we conducted comprehensive ablation studies across a diverse subset of datasets. We selected eight representative datasets spanning various machine learning tasks: yolanda [61], mldoc [64], flood [34], petfinder [2], camoseg [42], rvlcdip [27], solar(10m) [65], and fiqa [68]. We select this subset to include a broad spectrum of domains including tabular data analysis, multimodal data analysis, multi-lingual and multi-table tasks, document classification, image segmentation, time series forecasting, and text retrieval.

# B   Agents in MLZero

## B.1   Perception 👀

### B.1.1   File Grouping 📁 and File Perception 📖

Our file grouping mechanism (Algorithm 1) employs a hierarchical approach that analyzes folder structures and file extensions to identify meaningful patterns within raw datasets. We first define max group size $\delta$ ($\delta = 5$ by default in all our experiments). For each directory level, the algorithm dynamically determines whether to preserve specific folder names (when $\leq \delta$ unique names exist at that level) or abstract them using wildcards (when $> \delta$ names exist), creating a balanced representation that captures essential structural information while avoiding over-specification.

Upon establishing these file groups, our perception system selects representative examples for detailed analysis. For small groups $\leq \delta$, all members undergo comprehensive content inspection, while larger collections are represented by carefully selected exemplars. Each file is then processed through File Perception Agent, which dynamically generates format-appropriate loading and printing code based on file characteristics, thereby providing structured insights into raw data content regardless of format or organization. This two-phase perception approach forms the foundation of our system's ability to understand arbitrary data structures without manual preprocessing.

---

**Algorithm 1** File Grouping

---

1:  **procedure** FILE GROUPING(files)
2:      $depthFolders \leftarrow$ Map from depth to set of folders
                                                    ▷ First pass: analyze folder structure
3:      **for all** $file \in files$ **do**
4:          $paths \leftarrow$ SplitPath($file.path$)
5:          **for** $depth \leftarrow 0$ **to** $|paths| - 2$ **do**
6:              $depthFolders[depth].add(paths[depth])$
7:          **end for**
8:      **end for**
                                                    ▷ Second pass: group files
9:      $groups \leftarrow$ Map from pattern to file list
10:     **for all** $file \in files$ **do**
11:         $paths \leftarrow$ SplitPath($file.path$)
12:         $pattern \leftarrow []$
                                                    ▷ Build pattern using folder structure
13:         **for** $depth \leftarrow 0$ **to** $|paths| - 2$ **do**
14:             **if** $|depthFolders[depth]| \leq \delta$ **then**
15:                 $pattern.append(paths[depth])$          ▷ Use actual folder name
16:             **else**
17:                 $pattern.append("*")$                    ▷ Use wildcard
18:             **end if**
19:         **end for**
20:         $pattern.append(\text{GetExtension}(file.name))$
21:         $groups[pattern].append(file)$
22:     **end for**
23:     **return** $groups$
24: **end procedure**

---

> **File Perception Agent**
>
> **Context:** The agent utilizes LLM capabilities to analyze file content across various formats, enabling raw data understanding in AutoML workflows. This addresses a critical limitation in previous systems that required pre-structured data.
> **Prompt:**
> ```
> f"""Generate Python code to read and analyze the file: "{file_path}"
> ```

```
Your code should:
1. Import all modules used (e.g. import os).
2. Use appropriate libraries based on file type (pandas for tabular
data, etc.)
3. For tabular files (csv, excel, parquet, etc.):
   - Display column names. If there are more than 20 columns, only
   display the first and last 10.
   - Show first 2-3 rows with truncated cell content
   - Do not show additional index column if it's not in the original
   table
   - If failed to open the file, treat it as text file
    {"- Count total rows and provide basic statistics"
         if self.details else "- No additional info needed."}
4. For text files:
   - Display first few lines (up to {max_chars} characters)
5. For compressed tabular or text files,
     show its decompressed content as described.
{"6. For other files, provide appropriate summary"
     if self.details else
         "6. For binary or other files, provide only file size."}
7. Keep the total output under {max_chars} characters

Return ONLY the Python code, no explanations or markdown.
The code should be self-contained and executable on its own."""
```

**Generated Code Example:**

```python
import pandas as pd
import os

# Get absolute path and file size
file_path = "/media/agent/maab/datasets/abalone/training/test.csv"
abs_path = os.path.abspath(file_path)
file_size = os.path.getsize(file_path) / (1024 * 1024)

print(f"File Size: {file_size:.2f} MB")

# Read and analyze CSV file
try:
    df = pd.read_csv(file_path)

    # Display column names
    print(f"Column names: {list(df.columns)}")

    # Show first rows
    print("First rows:")
    print(df.head(3))

except Exception as e:
    print(f"Error reading file as CSV: {e}")

    # Try as text file if CSV reading fails
    try:
        with open(file_path, 'r') as f:
            text = f.read(1000)
        print("File content (first 1000 chars):")
        print(text)
    except Exception as e2:
```

```
          print(f"Error reading as text: {e2}")
```

**Example Output:**

```
Absolute path: /media/agent/maab/datasets/abalone/training/test.csv
Content:
File Size: 0.02 MB
Column names: ['Sex', 'Length', 'Diameter', 'Height',
'Whole_weight',  'Shucked_weight', 'Viscera_weight',
'Shell_weight', 'Class_number_of_rings']
First rows:
  Sex  Length  Diameter  Height  Whole_weight  Shucked_weight...
0   I   0.620     0.485    0.18        1.1785          0.4675...
1   F   0.645     0.525    0.17        1.3700          0.6135...
2   F   0.620     0.480    0.17        1.1045          0.5350...
```

### B.1.2   Task Perception 💡

The task perception in MLZero occurs in two main phases. First, the system uses an LLM to identify potential description files from the data prompt, looking specifically for files like READMEs, documentation, or task definitions. The LLM's response is parsed to extract filenames and returns both the list of files and the analysis explanation. Second, the system generates a comprehensive task description by reading the content of the identified files and using another LLM call to extract key information. It processes the file contents along with the original data prompt and previous analysis to produce a structured description that includes the core data science objective, requirements, constraints, data sources, and success metrics. These steps create a foundation for understanding what the task requires before proceeding to ML library selection.

---

**Task Perception Agent**

**Prompt to Find Description File**
```
f"""Given this data prompt:

{data_prompt}

Please identify any files that appear to contain project
descriptions, requirements, or task definitions.
Look for files like README, documentation files, or task
description files.

Format your response as follows:
Description Files: [list ONLY the absolute path, one per line]
Explanation: [explain why these files were identified as
description files]"""
```
**Prompt to Generate Task Descriptions**
```
f"""Based on this data prompt and description files:

Data Prompt:
(IMPORTANT: The metadata of example files in Data Prompt may not be
representative - do not make assumptions about data statistics based
on examples.)

{data_prompt}

Description File Analysis:
{description_analysis}

Description File Contents:
```

---

```
{description_context}

Based ONLY on the information explicitly stated in the provided data
prompt, description files, and analysis, provide a condensed
description of the data science task. Include only details that are
directly mentioned in the source materials.
Do not add assumptions or infer unstated information.
"""
```

### B.1.3 ML Library Selection 🛠️

The ML library selection process follows the task perception steps and uses an LLM to identify the most suitable ML library for the data science task at hand. It takes the data prompt, task description, and available libraries from the registry as inputs. The prompt includes the task details and formatted information about each available library (name, version, description, and special features). The LLM's response includes both the selected library name and the reasoning behind the selection.

---

**ML Library Selection Agent**

**Prompt to Select ML Libraries**

```
f"""Given the following data science task:

Data Description:
{data_prompt}

Task Analysis:
{description}

Available tools and their capabilities:

{_format_tools_info(tools_info)}

Please select the most appropriate tool for this task. Consider:
1. The nature of the data (tabular, time series, multimodal, etc.)
2. The specific requirements of the task
3. Any limitations or special features of each tool

Format your response as follows:
Selected Tool: [tool name ONLY]
Explanation: [detailed explanation of why this tool is the best
choice, including specific features that match the task
requirements]"""
```

---

## B.2 Semantic Memory 📚

### B.2.1 Condensation ✏️

The condensation agent processes tutorial content by breaking it into manageable chunks when necessary, then uses an LLM to condense each chunk while preserving essential implementation details, code samples, key concepts, critical configurations, and important warnings. It works on a chunk-by-chunk basis to handle large tutorials, maintains the overall document structure, and ensures the resulting condensed content stays within a specified maximum length by truncating at section boundaries when needed.

**Condensation Agent**

**Prompt to Condense a Document Chunk**

```
context = (
           "This is a continuation of the previous chunk. "
               if i > 0 else ""
          )
f"""{context}Condense this portion of the tutorial while preserving
essential implementation details, code samples, and key concepts.
Focus on:

1. Implementation details and techniques
2. Code snippets with necessary context
3. Critical configurations and parameters
4. Important warnings and best practices

Chunk {i+1}/{len(chunks)}:
{chunk}

Provide the condensed content in markdown format."""
```

### B.2.2  Summarization ⭐

The summarization agent generates concise summaries of the documents that help coder agent understand the implementation knowledge, coding tasks, and key features covered in the tutorial. It processes the already condensed content through an LLM to create a single paragraph summary of less than 100 words that starts with "Summary: " and highlights the most important aspects of the tutorial for practical implementation.

**Summarization Agent**

**Prompt to Summarize the Document**

```
f"""Generate a concise summary (within 100 words) of this tutorial
that helps a code generation LLM understand:
1. What specific implementation knowledge or techniques it can find
in this tutorial
2. What coding tasks this tutorial can help with
3. Key features or functionalities covered

Tutorial content:
{condensed_content}

Provide the summary in a single paragraph starting with
"Summary: "."""
```

### B.2.3  Retrieval 🔍

The retrieval agent scans documents of the selected ML libraries, extracts their titles and summaries, and uses an LLM to select the most relevant documents based on the user's task, data description, question, and any previous errors. The selected documents are then provided to coder agent to correctly integrate the selected ML library.

**Retrieval Agent**

**Prompt to Retrieve the Condensed Document**

```
context = f"""Task: {task_prompt}
```

```
Data: {data_prompt}
User Question: {user_prompt}
Previous Error: {error_prompt}"""

f"""Given the following context and list of tutorials with
their summaries, select the {max_num_tutorials} most relevant
tutorials for helping with this task. Consider how well each
tutorial's title and summary match the task, data, user
question, and any errors.

Context:
{context}

Available Tutorials:
{tutorials_info}

IMPORTANT: Respond ONLY with the numbers of the selected tutorials
(up to {max_num_tutorials}) separated by commas.
For example: "1,3,4" or "2,5" or just "1" if only one is relevant.
DO NOT include any other text, explanation, or formatting in
your response."""
```

## B.3  Episodic Memory ⓘ

### B.3.1  Error Analyzer ⚠

The error analyzer processes error messages from failed code executions. It takes the original task description, data, user prompt, previous code, and error message as input, and uses an LLM to generate a concise error summary to identify root causes and suggested debugging steps to provide tactical guidance. The agent is asked not to include actual code fixes, but focusing on clear, actionable insights within strict length constraints.

---

**Retrieval Agent**

**Prompt to Analyze the Error**

```
"""{task_prompt}
{data_prompt}
{user_prompt}
Previous Python Code:
{python_code}
Previous Bash Script to Execute the Python Code:
{bash_script}
{retrieved_tutorials}
Error Message:
{error_message}
Analyze the error message and context provided. Your response
MUST contain exactly two short paragraphs as follows:

ERROR SUMMARY: Provide a brief, technical description of the error
in 1-3 sentences. Focus only on identifying the root cause and
affected component without background explanations.

SUGGESTED FIX: Offer specific debugging directions in 1-3 sentences.
Do not include actual code or commands, only tactical debugging
guidance.
```

---

```
Each paragraph must be concise (maximum 3 sentences). Do not include
general advice, explanations beyond the direct debugging strategy, or
any additional paragraphs."""
```

## B.4   Iterative Coding 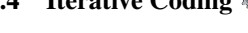

### B.4.1   Coder 👩‍💻

The Coder agent creates programming solutions based on user prompts. It supports both single-turn and multi-turn interactions. In our experiments, we use single-turn interaction by default as episodic memory can provide information about past errors. In the ablation study of removing episodic memory, we use multi-turn interactions where coders in all iterations run in a common session. The Code agent also ensures code is properly formatted within language-specific code blocks (python or bash in our experiments), extracting clean scripts from LLM responses.

For Python code generation, the prompt includes system prompt, library-specific instructions, task perception, data perception, optional user input, error analysis from episodic memory, and retrieved documents from semantic memory.

Similarly, the bash script generation prompt assembles contextual components including environment configuration preferences (tailored to whether environment setup is required), code execution instructions, the generated Python code to be run, and (when applicable) previous bash scripts and error messages.

---

**Coder Agent**

**Prompt to Generate Python Code**

```
f"""
As an AutoML Agent, you will be given a folder containing data and
description files. Please generate Python code using {tool_name} to
train a predictor and make predictions on test data. Follow these
specifications:

ONLY save files to the working directory: {output_folder}.

1. Data preprocessing:
   - Remove training data samples without valid labels (unless told
   not to do so).
   - Remove the unneccesary index column (if applicable)

2. Model training:
   - Use {tool_name} with appropriate parameters for the task
   - If a model is trained, save it in a folder with random timestamp
   within {output_folder}

3. Prediction:
   - Make predictions on the test data
   - Save the predicted results to {output_folder}, result file name
   should be "results", the format and extension should be same as
   the test data file
   - Output column names must exactly match those in the training or
   sample submission files without adding "predicted_" prefixes or
   creating any new columns.

4. Documentation:
   - Add a brief docstring at the beginning of the script explaining
   its purpose and usage
```

---

```
        - Also include additional installation steps with comments at the
        beginning of the script
        - Include comments explaining any complex operations or design
        decisions

    5. Others:
        - To avoid DDP errors, wrap the code in: if __name__ == "__main__":
        - Ensure errors are propagated up and not silently caught - do not
        use try/except blocks unless you explicitly reraise the exception.

    {tool_prompt}

    Please provide the complete Python script that accomplishes these
    tasks, ensuring it's ready to run given the appropriate data inputs.

    Task Description: {task_description}

    {data perception}

    {user input (optional)}

    {error analysis (from episodic memory)}

    {retrieved documents (from semantic memory)}
    """
```

**Code To Generate Prompt for Bash Script Generation**

```python
    # Truncate error message if needed
    if len(error_message) > max_error_message_length:
        error_message = (
            error_message[: max_error_message_length // 2]
            + "\n...(truncated)\n"
            + error_message[-max_error_message_length // 2 :]
        )

    # Build the core instructions
    instructions = []
    if create_venv:
        instructions.extend(
            [
                f"Create and configure a conda environment in
                {output_folder}:",
                "- Python version: 3.11",
                "- Activate the environment",
                "- Install required packages",
            ]
        )
    elif install_packages:
        instructions.append(
            "The environment may not be fully configured. Install any
            packages required in the python code."
        )
    else:
        instructions.append(
            "The environment is already configured. Do not install or
            update any package."
        )
```

```python
    instructions.append(f"Execute the Python script:
{python_file_path}")

    # Build the prompt with optional context
    prompt_parts = [
        "Generate a minimal bash script that will:",
        "\n".join(f"{i+1}. {instr}" for i, instr in
        enumerate(instructions)),
    ]

    if current_python:
        prompt_parts.append(
            dedent(
                f"""
            Current Python code:
            ```python
            {current_python}
            ```
        """
            ).strip()
        )

    if error_message:
        prompt_parts.append(f"Previous error:\n{error_message}")

    if previous_bash and error_message:
        prompt_parts.append(
            dedent(
                f"""
            Previous failed bash script:
            ```bash
            {previous_bash}
            ```
        """
            ).strip()
        )

    if previous_python and error_message:
        prompt_parts.append(
            dedent(
                f"""
            Previous Python code:
            ```python
            {previous_python}
            ```
        """
            ).strip()
        )

    # Add final instructions
    prompt_parts.append(
        dedent(
            """
        Notes:
        - Generate a minimal, executable bash script
        - Focus on essential commands only
        - Handle common environment and package only if there were
```

```
        errors
    """
        ).strip()
    )

    return "\n\n".join(prompt_parts)
```

## B.4.2 Executer 🚀

The Executer agent executes generated code and captures real-time outputs with timeout protection. Success determination is two-fold: first checking if the script execution completes with a zero return code, then passing execution logs to LLM that evaluates deeper success criteria beyond just exit codes. The Executer agent makes critical "FINISH" or "FIX" decisions with explainations based on log analysis, even overriding apparent successes when it detects logical errors or poor performance in the outputs, ensuring both technical completion and task fulfillment before concluding the execution cycle.

---

**Executer Agent**

**Prompt to Determine if Execution is Successful**

"""You are an expert code evaluator. Analyze the execution results of the following Python code and determine if the execution was successful or if issues need to be fixed.

{task_prompt}{data_prompt}

## Python Code
```python
{python_code}
```

## Execution Results
### Standard Output (stdout)
```
{stdout or "No standard output"}
```

### Standard Error (stderr)
```
{stderr or "No standard error"}
```

Evaluate the execution results and decide on one of the following actions:
1. FINISH - If the execution was completely successful and met all requirements.
2. FIX - If there were errors, issues, or performance problems that need to be addressed.

Provide your decision in the following format:
DECISION: [FINISH or FIX]
ANALYSIS: [Brief analysis of errors if any, or "None" if no errors]

The error analysis should be brief but informative enough for another agent to understand what needs to be fixed.

Even if the code executed without throwing errors, it might still

---

```
have issues with logic or not meet all requirements."""
```

# C  More Implementation and Competitors Details

## C.1  Implementation Details

For MLZero, the code execution environment operates with a 3-hour timeout and a maximum of 5 coding iterations. We enforce strict prompt size constraints: 1,024 characters for files, 8,192 characters for tutorials, and 2,048 characters for user inputs. We leverage Claude 3.7 Sonnet [6] as our LLM backbone, using temperature 0 for planning and file reading tasks, while employing temperature 0.5 for coding. All agents utilize a 65536-token context window with task-specific configurations. For the 8B configuration of MLZero, we adjusted parameters to accommodate the smaller context window. Specifically, we set the context window to 8,192 tokens across all agents, reduced retrieval size to 3, and limited maximum tutorial length to 4,069 characters.

To ensure fair comparison with AIDE's default setting (def), we standardized its configuration by disabling automatic file copying and utilizing identical LLM backbones [6] for code generation and feedback processes. We maintained a 3-hour timeout and 5-step reasoning process to match our method's 5 coding iterations. Additionally, we also conducted evaluations on an enhanced version (AIDE +ext) that incorporates six comprehensive tutorials covering fundamental knowledge for all tasks to facilitate coding with external ML libraries. In the enhanced configuration, we also provide the system prompt of MLZero to optimize AIDE's performance.

Codex CLI [57] performs end-to-end with minimal human intervention. We similarly use a 3-hour timeout but **no step limit**. For other comparative systems that typically require manual data pre-processing [33, 24, 45], we enhanced their capabilities by augmenting them with outputs from our data perception module, providing additional data insights that enable end-to-end execution on our benchmark. Since DS-Agent [24] lacks native code execution support, we manually executed their generated code to obtain results; thus relative execution times are unavailable as they would not represent true end-to-end agent performance. As both AIDE [38] and AutoKaggle [45] (AK) lack robust mechanisms for saving results through user instructions, we manually extracted results from their working directories for evaluation. We also standardized all description file names to accommodate baseline agents. For agents (Codex CLI, DS-Agent, AutoKaggle) that lack compatibility with Claude 3.7 Sonnet [6] due to implementation constraints or outdated interfaces, we substitute with GPT-4o [3], a general-purpose model with comparable benchmark performance.

As MLE-bench offers a longer time (24 hours) for each agent on each dataset, we prompt the ML libraries [19, 65, 67] to use best quality preset with a 3-hours time limit per iteration for each dataset. Other implementations in MLZero remain the same. For the implementation of AIDE, MLAgentBench [33], and OpenHands [72], please refer to MLE-bench [10].

All experiments were conducted on an AWS EC2 p4d.24xlarge instance equipped with 8 NVIDIA A100 40G GPUs. Computation time varied based on iterations required to achieve success. For the Multimodal AutoML Agent Benchmark, the maximum computational cost for running one agent is 3 hours $\times$ 25 datasets $\times$ 8 GPUs, totaling 600 GPU hours. The total computational upper bound for Table 1 is 600 GPU hours $\times$ 11 agents $\times$ 3 runs, totaling 19,800 GPU hours. For MLE-bench, the maximum computational cost for running MLZero once is 24 hours $\times$ 21 datasets $\times$ 8 GPUs, totaling 4,032 GPU hours.

The operational cost of using Claude 3.7 Sonnet varies with the number of iterations required for success. Approximately, MLZero incurs a cost of $0.25 per dataset, while AIDE averages $0.5 per dataset. However, these figures may fluctuate significantly depending on the iterations needed to achieve successful completion.

## C.2  MLZero

The default configuration of MLZero is attached below. Please check Appendix B for the detailed prompts and implementation of each agent in MLZero, and Appendix D for the ML libraries used.

**MLZero Default (def) Configuration**

```
stream_output: True
per_execution_timeout: 10800

max_chars_per_file: 1024
max_num_tutorials: 5
max_user_input_length: 2048
max_error_message_length: 2048
max_tutorial_length: 8192
create_venv: false
condense_tutorials: True

# Default LLM Configuration
# For each agent (coder, etc.) you can use a different one
llm: &default_llm
  # Note: bedrock is only supported in limited AWS regions
  #       and requires AWS credentials
  provider: bedrock
  model: "us.anthropic.claude-3-7-sonnet-20250219-v1:0"
  max_tokens: 65536
  proxy_url: null
  temperature: 0
  verbose: True
  multi_turn: False

coder:
  <<: *default_llm
  temperature: 0.5
  top_p: 1

planner:
  <<: *default_llm
  max_stdout_length: 8192
  max_stderr_length: 2048

file_reader:
  <<: *default_llm
  details: False
```

## C.3    Codex CLI

OpenAI's Codex CLI represents a practical implementation of LLM-based tool-using agents for software development. When equipped with strong reasoning models, it demonstrates effective instruction-following capabilities and can reliably execute software engineering tasks. However, it tends to generate simplistic solutions that underperform on complex machine learning workflows. Moreover, its performance degrades substantially when deployed with general-purpose language models that lack robust reasoning capabilities, underscoring the limitations of current tool-using agents in efficiently handling the sophisticated requirements of end-to-end machine learning automation.

In our experiments (Section 4.2), we evaluate Codex CLI in two configurations. The default configuration (Codex CLI) uses OpenAI's GPT-4.1 [56], a general-purpose model, to ensure fair comparison with other agents that employ non-reasoning models. We also include an enhanced configuration (Codex CLI +rea) utilizing o4-mini [58], a reasoning model, to demonstrate Codex CLI's upper performance bound. Both configurations employ the following standardized prompt:

## C.4 AIDE

AIDE [38] conceptualizes machine learning engineering as a code optimization problem, employing tree search methodologies within the solution space. Similar approaches are found in SELA [14] and I-MCTS [46], though these frameworks focus primarily on tabular data and do not provide end-to-end solutions.

For MLE-bench [10] comparisons, we utilize the officially reported AIDE results. However, for our Multimodal AutoML Agent Benchmark, we implemented several configuration modifications to ensure fair and efficient comparison:

1. We disabled automatic data copying functionality, as several datasets in our benchmark are extremely large, making this process prohibitively time-consuming.

2. Archive unpacking functionality was disabled to preserve the original data structure integrity.

3. Report generation was disabled as it increases token and time usage without affecting performance metrics.

4. Both code generation and feedback mechanisms were changed to use Claude 3.7 Sonnet, consistent with our system's configuration.

To facilitate proper evaluation within our benchmark framework, we implemented additional hard-coded configurations: (1) Description files are automatically fed into AIDE, as the system cannot independently locate these files; (2) Output files are renamed from AIDE's default "submission.csv" to each dataset's required filename, as AIDE uses hardcoded output naming conventions that would otherwise impede proper evaluation.

For the default AIDE configuration (AIDE def), we incorporated the following additional prompt to ensure output format consistency and proper file referencing:

**Additional Prompt for AIDE (def)**

```
Output column names must exactly match those in the training or
sample submission files without adding "predicted_" prefixes or
creating any new columns.
Always use absolute file paths when your model prediction outputs
reference files.
```

For the enhanced AIDE configuration (AIDE +ext), we further augmented the system with our MLZero coder's prompt, ML library prompts (detailed in Appendix B and Appendix D), and several critical condensed tutorials from MLZero's knowledge base. This enhanced configuration represents our effort to provide AIDE with comparable knowledge resources for a more equitable performance comparison:

**Additional Prompt for AIDE (+ext)**

```
[MLZero coder prompt]
[MLZero ML library prompts]
[Condensed Tutorials]
```

## C.5 DS-Agent

DS-Agent [24] leverages LLMs with case-based reasoning (CBR) to solve machine learning problems, particularly excelling in comprehending task requirements and constructing machine learning pipelines. Unlike ResearchAgent [33] focused primarily on generating reasonable plans, DS-Agent implements a two-stage approach:

1. A comprehensive development stage utilizing the full CBR framework to capitalize on expert knowledge from Kaggle competitions and iteratively improve performance through feedback mechanisms.

2. A resource-efficient deployment stage with streamlined CBR to adapt previously successful solutions for direct code generation.

For benchmarking DS-Agent, we utilized the official records of the development stage. As DS-Agent lacks an end-to-end execution system, we modified the Multimodal AutoML Agent Benchmark to align with the data requirements [33], then incorporated MLZero's perception results for each dataset as commented paragraphs as the provided skeleton code.

For both default (def) and zero-shot evaluation settings, we followed the original implementation without modifications. And we copied and executed the generated code within the Multimodal AutoML Agent Benchmark environment. To ensure consistent evaluation criteria across all systems, we also supplemented the description files with additional standardized prompts to enforce consistent output formatting and behavior.

---

**Additional Prompt for DS-Agent**

```
ONLY save files to: "./".
Make predictions on the test data.
Save the predicted results to "./", result file name should be
"results", the format and extension should be same as the test
data file.
Output column names must exactly match those in the training or sample
submission files without adding "predicted_" prefixes or creating any
new columns.
Tensorflow is not installed. But you can use pytorch when needed.
```

---

## C.6 AutoKaggle

AutoKaggle [45] presents a collaborative multi-agent framework for automated machine learning solutions, comprising five specialized agents (Reader, Planner, Developer, Reviewer, and Summarizer) working in concert throughout the solution development lifecycle. The framework implements an iterative development methodology with comprehensive testing procedures and leverages a predefined machine learning tools library. However, a significant limitation is its rigid input data format requirements, as it exclusively supports evaluation of tabular datasets from Kaggle competitions within a predefined directory structure. For AutoKaggle to process data correctly, the input files must adhere to the following structural convention:

---

**Required Data Format for AutoKaggle**

```
competition/
 - train.csv
 - test.csv
 - sample_submission.csv
 - overview.txt
```

---

To accommodate these constraints while evaluating AutoKaggle within our benchmark framework, we implemented several adaptations to preprocess data: renaming `descriptions.txt` files to `overview.txt`; augmenting training and test datasets with sequential `index` columns; converting diverse data formats to the mandatory CSV format with standardized `train.csv` and `test.csv`

filenames; and generating appropriate `sample_submission.csv` files that conform to the expected index and target column structure. These adaptations ensure fair evaluation while maintaining the integrity of the original datasets within our comparative benchmark.

# D ML Libraries in MLZero

Our system aims to minimize required human effort by leveraging ML frameworks with comprehensive capabilities rather than specialized libraries. Each integrated framework may contain undocumented issues (mentioned in GitHub issues rather than official documentation), for which we provide supplementary prompts to enhance LLM performance. These prompts were also provided to AIDE (+ext) to ensure fair comparison. The MLZero (-ext) configuration include only general descriptions of machine learning algorithms D.5.

MLZero incorporates only a few well-maintained ML frameworks that demonstrate state-of-the-art performance across various modalities, keeping the ML libraries used minimal to avoid "overfitting" the tasks. Our selection criteria prioritize frameworks that provide comprehensive automation capabilities, robust implementation quality, and consistent maintenance to ensure reliable performance. This approach enables our system to handle a wide spectrum of machine learning tasks with minimal human intervention while maintaining competitive performance. Adding more ML libraries with better performance could further improve the agent's performance, but we intentionally limited our selection to prevent overfitting to our evaluation benchmarks.

## D.1 Tabular

For tabular data processing, we integrate frameworks with strong performance on structured data tasks [19, 40]. The time_limit and presets parameters are specified for benchmarking purposes but can be adjusted through user input in practical applications.

---

**Tool Registry for Tabular Tasks**

```
{
  "name": "autogluon.tabular",
  "version": "1.2.0",
  "description": "AutoGluon Tabular is an open-
source AutoML framework
  that automates the training and tuning of machine learning models
  for tabular data, handling tasks from preprocessing to model
  ensembling with minimal code required.",
  "features": [
    "Works best when there are only tabular data (categorical and
    numerical).",
    "Does not work very well on nlp tasks.",
    "Does not work with image data."
  ],
  "requirements": [],
  "prompt_template": [
    "Use Autogluon Tabular with the following parameters:",
    "- time_limit: 1800 seconds",
    "- presets: \\\"medium_quality\\\"",
    "- tuning_data: only use validation if there is a validation
    dataset.",
    "- problem_type: binary, multiclass, or regression."
  ]
}
```

## D.2 Multimodal

For multimodal tasks, we selected AutoMM [67] due to its performance in multimodal AutoML tasks and comprehensive support across diverse problem types. This framework excels at integrating multiple data modalities into unified representations, providing good performance with minimal configuration requirements.

---

**Tool Registry for Multimodal Tasks**

```
{
  "name": "autogluon.multimodal",
  "version": "1.2.0",
  "description": "AutoGluon Multimodal is an open-source AutoML
  framework that simplifies the training of models across multiple data
  types including text, images, and tabular data, automating tasks from
  preprocessing to model ensembling with minimal code required.",
  "features": [
    "Support multimodal classification or regression, document
    classification, semantic segmentation",
    "Does not work the best with pure tabular data (categorical and
    numerical).",
    "Does not support image or text generation tasks."
  ],
  "requirements": [],
  "prompt_template": [
    "Use Autogluon Multimodal with the following parameters:",
    "- time_limit: 1800 seconds",
    "- presets: \\\"medium_quality\\\"",
    "- tuning_data: only use validation if there is a validation
    dataset.",
    "The usage of document prediction is different from image
    prediction.",
    "Check data path carefully when encounter ValueError: No model is
    available for this dataset.",
    "For semantic segmentation, use single GPU by setting
    CUDA_VISIBLE_DEVICES=0",
    "For semantic segmentation, save the mask as greyscale JPG image
    (squeeze then cv2.imwrite) in \\\"predicted_mask\\\" folder under
    output folder and save its absolute path in label column.",
    "No need to specify model.names, and do not increase default per gpu
    batch size to avoid OOM errors.",
  ]
}
```

---

The selection of AutoMM aligns with our approach of leveraging well-maintained frameworks with comprehensive capabilities. Its performance across diverse problem types (image-text classification, document understanding, semantic segmentation, etc.) enables MLZero to handle a wide spectrum of multimodal tasks without requiring specialized implementations for each modality combination. In future work, more advanced multimodal ML libraries should be included to support additional problem types and to further improve performance.

## D.3 Timeseries

For timeseries forecasting, we incorporate the ML library with state-of-the-art performance [65]. Note that [19, 67, 65] share a similar API but internal logics are completely different and thus used as three libraries.

**Tool Registry for TimeSeries Tasks**

```
{
  "name": "autogluon.timeseries",
  "version": "1.2.0",
  "description": "AutoGluon Timeseries is an open-source AutoML
  framework that automates the training and tuning of forecasting models
  for time series data, handling tasks from preprocessing to model
  ensembling with built-in support for both univariate and multivariate
  forecasting.",
  "features": [
    "timeseries forecasting"
  ],
  "requirements": [],
  "prompt_template": [
    "DO NOT drop any data samples (to make sure the frequency
    isregular).",
    "Use Autogluon Timeseries with the following parameters:",
    "- time_limit: 1800 seconds",
    "- presets: \\\"medium_quality\\\"",
    "  - tuning_data: only use validation if there is a
    validationdataset.",
    "Note that the prediction is given in a column named \"mean\". You
    need to rename the column in the result.",
    "'from_data_frame()' method of TimeSeriesDataFrame does not accept a
    'target' parameter.",
    "If there are known covariates, they should be specified in both
    TimeSeriesPredictor initialization AND predict."
  ]
}
```

## D.4 Retrieval

Our system incorporates FlagEmbedding [74, 80] to support tasks requiring semantic search, document ranking, and retrieval-augmented generation capabilities.

**Tool Registry for Retrieval Tasks**

```
{
  "name": "FlagEmbedding",
  "version": "1.3.4",
  "description": "Retrieval and Retrieval-augmented LLMs",
  "features": [
    "retrieval",
    "reranking"
  ],
  "requirements": [],
  "prompt_template": [
    "DO NOT SAVE THE MODEL."
  ]
}
```

## D.5 Others

When specialized frameworks are not suitable, e.g. image-to-image generation, audio tasks, sequence-to-sequence generation, etc., MLZero can fall back to general machine learning algorithms, providing greater flexibility for diverse tasks.

```
{
  "name": "machine learning",
  "version": "0.1.0",
  "description": "You should select this as a general reference of
  machine learning or deep learning algorithms in case other tools are
  not helpful.",
  "features": [],
  "requirements": [],
  "prompt_template": [
    "In the bash script, install all necessary packages."
  ]
}
```

## D.6 Library Integration Process

One of the key design principles of MLZero is the ease of integrating new ML libraries into the system's knowledge base. Adding a new library requires approximately one minute of human effort using our provided registration Python script. Users provide only four basic pieces of information: the library name (e.g., "FlagEmbedding"), version number (e.g., "FlagEmbedding==0.x.x"), a one-sentence description, and the path to the directory containing the library's official documentation. Users may optionally provide additional prompts or usage guidelines, though this is not required. Importantly, no coding skills are needed for this registration process.

After this brief human setup, the system operates completely autonomously. The Summarization and Condensation agents automatically process the provided documentation into structured, condensed formats suitable for the Semantic Memory module. This automated processing includes extracting relevant sections from official documentation, using LLM agents to condense lengthy tutorials into essential information, organizing information by topics and APIs, and building retrieval indices for efficient semantic search. While this processing requires computational time (typically 10-30 minutes depending on documentation size), it requires zero human intervention or supervision.

Once a library is successfully integrated into Semantic Memory, MLZero operates from raw data to final predictions without further human input. This stands in stark contrast to many existing AutoML systems that require manual preprocessing, hard-coded logic tailored to individual datasets, expert configuration of hyperparameters, or post-processing to extract results. Our experiments demonstrate that this design achieves true end-to-end automation while maintaining extensibility across diverse modalities and problem types.

# E Detailed Results

## E.1 Detailed Results on Multimodal AutoML Agent Benchmark

This appendix provides a detailed breakdown of all experimental runs conducted with different agents and configurations on the Multimodal AutoML Agent Benchmark in Section 4.2. Each table presents the complete results across three independent runs for each dataset in the benchmark suite: **MLZero def** in Table 7, **MLZero 8B** in Table 8, **MLZero -ext** in Table 9, **MLZero -epi** in Table 10, **Codex CLI def** in Table 11, **Codex CLI +rea** in Table 12, **AIDE def** in Table 13, **AIDE +ext** in Table 14, **DS-Agent def** in Table 15, **DS-Agent zeroshot** in Table 16, and **AutoKaggle def** in Table 17.

For each run, we report the performance metric (see Appendix A.1) and the computational time in seconds required to complete the task. The symbol $\times$ indicates runs where the model failed to complete or with invalid outputs. These detailed results complement the aggregated performance metrics discussed in the main paper and demonstrate the consistency and robustness of our approach across multiple executions.

Table 7: Three runs of **MLZero (def)** on Multimodal AutoML Agent Benchmark.

| Dataset | Run 1 | | Run 2 | | Run 3 | |
|---|---|---|---|---|---|---|
| | Result | Time (s) | Result | Time (s) | Result | Time (s) |
| abalone | 2.135 | 142 | 2.120 | 111 | 2.135 | 112 |
| airbnb_melbourne | 0.426 | 1202 | 0.428 | 998 | 0.426 | 1041 |
| airlines | 0.656 | 1285 | 0.658 | 442 | 0.656 | 1262 |
| bioresponse | 0.801 | 3050 | 0.814 | 497 | 0.801 | 167 |
| camo_sem_seg | 0.833 | 4089 | 0.839 | 2192 | 0.843 | 6358 |
| cd18 | 0.445 | 215 | 0.495 | 501 | 0.445 | 261 |
| climate_fever | 0.476 | 649 | 0.471 | 342 | 0.479 | 3719 |
| covertype | 0.976 | 1963 | 0.976 | 2634 | 0.976 | 4556 |
| electricity_hourly | -1.436 | 875 | -1.427 | 1848 | -1.393 | 2779 |
| europeanflooddepth | 0.680 | 524 | 0.690 | 615 | 0.690 | 509 |
| fiqabeir | 0.482 | 370 | 0.507 | 416 | 0.505 | 889 |
| gnad10 | 0.837 | 717 | 0.911 | 725 | 0.829 | 722 |
| ham10000 | 0.555 | 815 | 0.555 | 771 | 0.779 | 1049 |
| hateful_meme | 0.571 | 682 | 0.587 | 663 | 0.600 | 628 |
| isic2017 | 0.751 | 4139 | × | × | 0.758 | 8440 |
| kick_starter_funding | 0.438 | 1330 | 0.471 | 770 | 0.435 | 1027 |
| memotion | 0.503 | 3852 | 0.483 | 4460 | 0.514 | 3807 |
| mldoc | 0.956 | 3699 | 0.949 | 3588 | 0.948 | 2836 |
| nn5_daily_without_missing | -0.765 | 251 | -0.765 | 485 | × | × |
| petfinder | 0.387 | 701 | 0.387 | 747 | 0.397 | 791 |
| road_segmentation | 0.466 | 2069 | × | × | × | × |
| rvl_cdip | 0.871 | 2979 | × | × | 0.872 | 2868 |
| solar_10_minutes | -2.273 | 800 | -0.704 | 666 | × | × |
| women_clothing_review | 0.748 | 958 | 0.748 | 1068 | 0.751 | 935 |
| yolanda | 8.533 | 2066 | 8.533 | 2007 | 8.533 | 1902 |

Table 8: Three runs of **MLZero (8B)** on Multimodal AutoML Agent Benchmark.

| Dataset | Run 1 | | Run 2 | | Run 3 | |
|---|---|---|---|---|---|---|
| | Result | Time (s) | Result | Time (s) | Result | Time (s) |
| abalone | 2.087 | 385 | 2.087 | 353 | 2.087 | 374 |
| airbnb_melbourne | 0.428 | 3857 | 0.414 | 3054 | × | × |
| airlines | 0.731 | 3939 | × | × | 0.656 | 6638 |
| bioresponse | × | × | 0.797 | 550 | 0.797 | 2113 |
| camo_sem_seg | × | × | × | × | × | × |
| cd18 | -0.857 | 1061 | 0.445 | 506 | × | × |
| climate_fever | × | × | × | × | × | × |
| covertype | 0.976 | 7634 | × | × | 0.974 | 8769 |
| electricity_hourly | × | × | × | × | × | × |
| europeanflooddepth | 0.695 | 738 | × | × | × | × |
| fiqabeir | × | × | × | × | × | × |
| gnad10 | 0.850 | 3453 | 0.843 | 2574 | 0.843 | 3451 |
| ham10000 | 0.569 | 1703 | 0.569 | 1716 | × | × |
| hateful_meme | 0.607 | 2060 | × | × | 0.531 | 5122 |
| isic2017 | × | × | × | × | × | × |
| kick_starter_funding | 0.327 | 3309 | 0.438 | 4400 | 0.466 | 3382 |
| memotion | × | × | × | × | × | × |
| mldoc | 0.947 | 4236 | 0.948 | 5422 | 0.949 | 7009 |
| nn5_daily_without_missing | × | × | × | × | × | × |
| petfinder | 0.405 | 3355 | × | × | 0.389 | 3224 |
| road_segmentation | × | × | × | × | × | × |
| rvl_cdip | × | × | × | × | × | × |
| solar_10_minutes | × | × | × | × | × | × |
| women_clothing_review | 0.724 | 5132 | × | × | 0.503 | 2337 |
| yolanda | 8.533 | 7263 | 8.540 | 8669 | 8.533 | 4141 |

Table 9: Three runs of **MLZero (-ext)** on Multimodal AutoML Agent Benchmark.

| Dataset | Run 1 | | Run 2 | | Run 3 | |
|---|---|---|---|---|---|---|
| | Result | Time (s) | Result | Time (s) | Result | Time (s) |
| abalone | 2.218 | 165 | 2.196 | 352 | 2.158 | 290 |
| airbnb_melbourne | 0.398 | 329 | 0.087 | 797 | × | × |
| airlines | 0.642 | 422 | 0.618 | 234 | 0.644 | 160 |
| bioresponse | 0.877 | 659 | 0.800 | 333 | 0.801 | 145 |
| camo_sem_seg | × | × | 0.464 | 813 | × | × |
| cd18 | -0.634 | 223 | -4.145 | 2934 | 0.078 | 277 |
| climate_fever | × | × | 0.231 | 849 | 0.253 | 2808 |
| covertype | 0.887 | 688 | 0.876 | 120 | × | × |
| electricity_hourly | 1.753 | 848 | × | × | 1.755 | 10422 |
| europeanflooddepth | 0.583 | 335 | 0.707 | 347 | 0.496 | 843 |
| fiqabeir | × | × | 0.225 | 4573 | × | × |
| gnad10 | 0.883 | 714 | 0.872 | 219 | 0.873 | 184 |
| ham10000 | 0.671 | 6045 | × | × | × | × |
| hateful_meme | 0.382 | 3521 | × | × | 0.310 | 638 |
| isic2017 | × | × | × | × | × | × |
| kick_starter_funding | 0.442 | 797 | 0.335 | 522 | 0.303 | 438 |
| memotion | × | × | 0.655 | 4394 | 0.999 | 1695 |
| mldoc | 0.961 | 1524 | 0.928 | 872 | × | × |
| nn5_daily_without_missing | × | × | 1.459 | 8994 | 0.823 | 165 |
| petfinder | 0.357 | 338 | 0.385 | 1803 | 0.393 | 3340 |
| road_segmentation | × | × | 0.444 | 10253 | 0.179 | 3866 |
| rvl_cdip | 0.886 | 4829 | × | × | × | × |
| solar_10_minutes | × | × | × | × | × | × |
| women_clothing_review | 0.617 | 225 | 0.659 | 390 | 0.697 | 809 |
| yolanda | 8.892 | 290 | 8.931 | 307 | 8.974 | 338 |

Table 10: Three runs of **MLZero (-epi)** on Multimodal AutoML Agent Benchmark.

| Dataset | Run 1 | | Run 2 | | Run 3 | |
|---|---|---|---|---|---|---|
| | Result | Time (s) | Result | Time (s) | Result | Time (s) |
| abalone | 2.135 | 115 | 2.120 | 107 | 2.135 | 265 |
| airbnb_melbourne | 0.423 | 3217 | 0.415 | 901 | 0.428 | 1086 |
| airlines | 0.656 | 1232 | 0.658 | 406 | 0.656 | 1215 |
| bioresponse | 0.797 | 178 | 0.801 | 174 | 0.797 | 197 |
| camo_sem_seg | 0.841 | 2239 | 0.835 | 4304 | 0.840 | 6320 |
| cd18 | 0.445 | 166 | 0.495 | 494 | 0.586 | 526 |
| climate_fever | × | × | 0.476 | 3001 | 0.475 | 6657 |
| covertype | 0.976 | 2652 | 0.976 | 1924 | 0.976 | 1948 |
| electricity_hourly | -1.406 | 848 | -1.396 | 1012 | -1.393 | 781 |
| europeanflooddepth | 0.680 | 504 | 0.680 | 541 | 0.690 | 475 |
| fiqabeir | 0.376 | 462 | 0.494 | 219 | 0.515 | 192 |
| gnad10 | 0.837 | 709 | 0.837 | 745 | 0.829 | 690 |
| ham10000 | × | × | 0.555 | 752 | 0.779 | 669 |
| hateful_meme | 0.571 | 648 | 0.587 | 653 | 0.600 | 608 |
| isic2017 | × | × | × | × | × | × |
| kick_starter_funding | 0.438 | 2437 | 0.438 | 742 | 0.438 | 1226 |
| memotion | × | × | 0.507 | 2696 | 0.514 | 2462 |
| mldoc | 0.948 | 1117 | 0.948 | 2970 | 0.946 | 4681 |
| nn5_daily_without_missing | -0.765 | 219 | -0.764 | 201 | -0.765 | 211 |
| petfinder | 0.387 | 714 | 0.387 | 743 | 0.397 | 757 |
| road_segmentation | × | × | 0.597 | 2047 | 0.601 | 4903 |
| rvl_cdip | 0.871 | 2461 | × | × | 0.871 | 2570 |
| solar_10_minutes | -1.286 | 871 | × | × | × | × |
| women_clothing_review | 0.748 | 1073 | 0.657 | 590 | 0.751 | 957 |
| yolanda | 8.533 | 1908 | 8.533 | 1983 | 8.533 | 2010 |

Table 11: Three runs of **Codex CLI (def)** on Multimodal AutoML Agent Benchmark.

| Dataset | Run 1 | | Run 2 | | Run 3 | |
|---|---|---|---|---|---|---|
| | Result | Time (s) | Result | Time (s) | Result | Time (s) |
| abalone | × | × | 2.230 | 47 | × | × |
| airbnb_melbourne | × | × | × | × | × | × |
| airlines | × | × | × | × | × | × |
| bioresponse | 0.792 | 30 | × | × | × | × |
| camo_sem_seg | × | × | × | × | × | × |
| cd18 | -0.412 | 85 | -1.477 | 173 | × | × |
| climate_fever | × | × | × | × | × | × |
| covertype | × | × | 0.955 | 48 | × | × |
| electricity_hourly | × | × | × | × | × | × |
| europeanflooddepth | × | × | × | × | × | × |
| fiqabeir | × | × | × | × | × | × |
| gnad10 | × | × | 0.790 | 111 | 0.847 | 159 |
| ham10000 | 0.484 | 36 | × | × | × | × |
| hateful_meme | × | × | × | × | × | × |
| isic2017 | × | × | × | × | × | × |
| kick_starter_funding | × | × | × | × | × | × |
| memotion | × | × | × | × | 0.532 | 92 |
| mldoc | × | × | 0.094 | 114 | 0.553 | 85 |
| nn5_daily_without_missing | × | × | × | × | × | × |
| petfinder | × | × | × | × | × | × |
| road_segmentation | × | × | × | × | × | × |
| rvl_cdip | × | × | × | × | × | × |
| solar_10_minutes | × | × | × | × | × | × |
| women_clothing_review | × | × | × | × | × | × |
| yolanda | × | × | × | × | × | × |

Table 12: Three runs of **Codex CLI (+rea)** on Multimodal AutoML Agent Benchmark.

| Dataset | Run 1 | | Run 2 | | Run 3 | |
|---|---|---|---|---|---|---|
| | Result | Time (s) | Result | Time (s) | Result | Time (s) |
| abalone | 2.232 | 201 | 2.232 | 93 | 2.351 | 299 |
| airbnb_melbourne | 0.384 | 85 | 0.388 | 108 | 0.384 | 86 |
| airlines | 0.653 | 175 | 0.612 | 177 | 0.628 | 81 |
| bioresponse | 0.882 | 163 | 0.800 | 90 | 0.880 | 117 |
| camo_sem_seg | × | × | × | 81 | × | 162 |
| cd18 | -1.288 | 84 | -1.685 | 136 | -1.362 | 125 |
| climate_fever | 0.123 | 1029 | 0.233 | 165 | 0.231 | 155 |
| covertype | 0.957 | 84 | 0.955 | 90 | 0.841 | 75 |
| electricity_hourly | × | × | × | 75 | × | 101 |
| europeanflooddepth | 0.435 | 219 | × | 190 | × | 100 |
| fiqabeir | 0.199 | 171 | × | 204 | 0.199 | 120 |
| gnad10 | 0.848 | 116 | 0.845 | 123 | 0.845 | 131 |
| ham10000 | 0.511 | 471 | 0.451 | 80 | 0.451 | 75 |
| hateful_meme | 0.572 | 119 | 0.454 | 110 | 0.403 | 61 |
| isic2017 | 0.111 | 473 | × | 440 | × | 391 |
| kick_starter_funding | 0.362 | 109 | 0.362 | 72 | 0.285 | 76 |
| memotion | × | × | 0.702 | 95 | 0.819 | 152 |
| mldoc | 0.933 | 221 | 0.788 | 144 | 0.739 | 172 |
| nn5_daily_without_missing | × | × | × | 166 | × | 149 |
| petfinder | 0.396 | 115 | 0.396 | 108 | 0.396 | 67 |
| road_segmentation | × | × | × | 131 | × | 154 |
| rvl_cdip | × | × | × | 128 | × | 147 |
| solar_10_minutes | × | × | × | 148 | -1.286 | 221 |
| women_clothing_review | 0.419 | 73 | 0.029 | 157 | 0.589 | 127 |
| yolanda | 9.091 | 135 | 9.598 | 86 | 9.598 | 75 |

Table 13: Three runs of **AIDE (def)** on Multimodal AutoML Agent Benchmark.

| Dataset | Run 1 | | Run 2 | | Run 3 | |
|---|---|---|---|---|---|---|
| | Result | Time (s) | Result | Time (s) | Result | Time (s) |
| abalone | 2.182 | 483 | 2.095 | 234 | 2.188 | 445 |
| airbnb_melbourne | × | × | 0.394 | 174 | × | × |
| airlines | × | × | × | × | × | × |
| bioresponse | 0.868 | 434 | 0.877 | 933 | × | × |
| camo_sem_seg | × | × | × | × | × | × |
| cd18 | × | × | × | × | × | × |
| climate_fever | × | × | × | × | × | × |
| covertype | × | × | × | × | 0.880 | 197 |
| electricity_hourly | × | × | × | × | × | × |
| europeanflooddepth | 0.722 | 2170 | × | × | 0.693 | 4365 |
| fiqabeir | × | × | × | × | × | × |
| gnad10 | × | × | 0.908 | 10812 | 0.897 | 2912 |
| ham10000 | 0.809 | 1492 | × | × | 0.812 | 3839 |
| hateful_meme | 0.557 | 2672 | 0.467 | 1907 | × | × |
| isic2017 | × | × | × | × | × | × |
| kick_starter_funding | × | × | × | × | × | × |
| memotion | 0.471 | 1050 | × | × | × | × |
| mldoc | × | × | 0.965 | 10801 | × | × |
| nn5_daily_without_missing | × | × | × | × | × | × |
| petfinder | × | × | × | × | 0.336 | 652 |
| road_segmentation | × | × | × | × | × | × |
| rvl_cdip | × | × | × | × | × | × |
| solar_10_minutes | -1.054 | 2698 | × | × | × | × |
| women_clothing_review | × | × | × | × | × | × |
| yolanda | × | × | × | × | × | × |

Table 14: Three runs of **AIDE (+ext)** on Multimodal AutoML Agent Benchmark.

| Dataset | Run 1 | | Run 2 | | Run 3 | |
|---|---|---|---|---|---|---|
| | Result | Time (s) | Result | Time (s) | Result | Time (s) |
| abalone | 2.189 | 385 | 2.135 | 651 | 2.230 | 469 |
| airbnb_melbourne | 0.415 | 2689 | × | × | 0.363 | 329 |
| airlines | × | × | 0.655 | 3771 | 0.643 | 155 |
| bioresponse | 0.876 | 7400 | 0.801 | 309 | × | × |
| camo_sem_seg | × | × | × | × | × | × |
| cd18 | 0.239 | 248 | × | × | -0.335 | 113 |
| climate_fever | × | × | × | × | × | × |
| covertype | 0.744 | 7856 | 0.974 | 7415 | 0.860 | 182 |
| electricity_hourly | × | × | × | × | × | × |
| europeanflooddepth | 0.697 | 730 | 0.680 | 3784 | 0.659 | 1654 |
| fiqabeir | × | × | × | × | × | × |
| gnad10 | × | × | × | × | 0.585 | 309 |
| ham10000 | 0.785 | 813 | 0.796 | 5001 | 0.847 | 8672 |
| hateful_meme | 0.518 | 2045 | 0.501 | 10802 | 0.442 | 637 |
| isic2017 | × | × | × | × | × | × |
| kick_starter_funding | 0.440 | 7496 | 0.438 | 1970 | × | × |
| memotion | × | × | × | × | × | × |
| mldoc | × | × | 0.948 | 1082 | 0.925 | 195 |
| nn5_daily_without_missing | × | × | × | × | × | × |
| petfinder | 0.382 | 2359 | 0.367 | 679 | 0.390 | 171 |
| road_segmentation | × | × | × | × | × | × |
| rvl_cdip | × | × | × | × | × | × |
| solar_10_minutes | × | × | × | × | × | × |
| women_clothing_review | 0.747 | 1134 | × | × | × | × |
| yolanda | 8.540 | 1953 | × | × | 9.032 | 172 |

Table 15: Three runs of **DS-Agent (def)** on Multimodal AutoML Agent Benchmark.

| Dataset | Run 1 | | Run 2 | | Run 3 | |
|---|---|---|---|---|---|---|
| | Result | Time (s) | Result | Time (s) | Result | Time (s) |
| abalone | 2.238 | 19 | × | × | × | × |
| airbnb_melbourne | × | × | × | × | × | × |
| airlines | × | × | × | × | × | × |
| bioresponse | 0.790 | 4 | 0.793 | 4 | × | × |
| camo_sem_seg | × | × | × | × | × | × |
| cd18 | -1.938 | 137 | × | × | × | × |
| climate_fever | × | × | × | × | × | × |
| covertype | 0.957 | 110 | × | × | × | × |
| electricity_hourly | × | × | × | × | × | × |
| europeanflooddepth | × | × | × | × | × | × |
| fiqabeir | × | × | × | × | × | × |
| gnad10 | × | × | × | × | × | × |
| ham10000 | × | × | × | × | × | × |
| hateful_meme | × | × | × | × | × | × |
| isic2017 | × | × | × | × | × | × |
| kick_starter_funding | × | × | × | × | × | × |
| memotion | × | × | × | × | × | × |
| mldoc | × | × | 0.949 | 1265 | × | × |
| nn5_daily_without_missing | × | × | -4.682 | 6 | × | × |
| petfinder | 0.355 | 9 | 0.350 | 3 | 0.369 | 4 |
| road_segmentation | × | × | × | × | × | × |
| rvl_cdip | × | × | × | × | × | × |
| solar_10_minutes | × | × | × | × | × | × |
| women_clothing_review | × | × | × | × | × | × |
| yolanda | × | × | × | × | × | × |

Table 16: Three runs of **DS-Agent (zero-shot)** on Multimodal AutoML Agent Benchmark.

| Dataset | Run 1 | | Run 2 | | Run 3 | |
|---|---|---|---|---|---|---|
| | Result | Time (s) | Result | Time (s) | Result | Time (s) |
| abalone | 2.362 | 2 | × | × | × | × |
| airbnb_melbourne | × | × | × | × | 0.314 | 16 |
| airlines | 0.612 | 61 | × | × | × | × |
| bioresponse | 0.793 | 4 | × | × | 0.793 | 4 |
| camo_sem_seg | × | × | × | × | × | × |
| cd18 | 0.338 | 2 | × | × | -1.615 | 3 |
| climate_fever | × | × | × | × | × | × |
| covertype | 0.952 | 99 | × | × | × | × |
| electricity_hourly | × | × | × | × | 11.662 | 10 |
| europeanflooddepth | × | × | × | × | × | × |
| fiqabeir | × | × | × | × | × | × |
| gnad10 | 0.691 | 14 | 0.903 | 304 | × | × |
| ham10000 | × | × | × | × | × | × |
| hateful_meme | × | × | × | × | × | × |
| isic2017 | × | × | × | × | × | × |
| kick_starter_funding | × | × | × | × | × | × |
| memotion | × | × | × | × | × | × |
| mldoc | × | × | × | × | × | × |
| nn5_daily_without_missing | × | × | × | × | × | × |
| petfinder | 0.389 | 4 | 0.214 | 8 | 0.207 | 12 |
| road_segmentation | × | × | × | × | × | × |
| rvl_cdip | × | × | × | × | × | × |
| solar_10_minutes | × | × | × | × | × | × |
| women_clothing_review | × | × | × | × | 0.353 | 18 |
| yolanda | × | × | × | × | × | × |

Table 17: Three runs of **AutoKaggle (def)** on Multimodal AutoML Agent Benchmark.

| Dataset | Run 1 | | Run 2 | | Run 3 | |
|---|---|---|---|---|---|---|
| | Result | Time (s) | Result | Time (s) | Result | Time (s) |
| abalone | × | × | × | × | × | × |
| airbnb_melbourne | 0.253 | 2886 | 0.314 | 2409 | 0.383 | 1165 |
| airlines | × | × | × | × | × | × |
| bioresponse | × | × | × | × | × | × |
| camo_sem_seg | × | × | × | × | × | × |
| cd18 | -1.840 | 1972 | × | × | × | × |
| climate_fever | × | × | × | × | × | × |
| covertype | × | × | 0.941 | 9284 | × | × |
| electricity_hourly | × | × | × | × | × | × |
| europeanflooddepth | × | × | × | × | 0.583 | 2017 |
| fiqabeir | × | × | × | × | × | × |
| gnad10 | × | × | 0.105 | 11281 | × | × |
| ham10000 | × | × | × | × | × | × |
| hateful_meme | 0.337 | 1732 | 0.382 | 1945 | × | × |
| isic2017 | × | × | × | × | × | × |
| kick_starter_funding | × | × | × | × | 0.237 | 2935 |
| memotion | × | × | × | × | × | × |
| mldoc | × | × | × | × | × | × |
| nn5_daily_without_missing | × | × | × | × | × | × |
| petfinder | 0.388 | 1881 | × | × | × | × |
| road_segmentation | × | × | × | × | × | × |
| rvl_cdip | × | × | × | × | × | × |
| solar_10_minutes | × | × | × | × | × | × |
| women_clothing_review | × | × | × | × | × | × |
| yolanda | × | × | × | × | × | × |

## E.2 Detailed Results on MLE-bench Lite

This appendix presents the comprehensive experimental results that supplement the analysis on MLE-bench Lite provided in the Section 4.2 of main paper. Table 18 offers a detailed comparison of our method against three state-of-the-art baseline approaches across all 21 datasets in the MLE-Bench Lite benchmark. Note that for metrics where lower values originally indicated better performance, we applied a negative sign to convert them, ensuring that higher values consistently represent better performance throughout the table. See Appendix A.2 for details about MLE-bench Lite. Gold, silver, and bronze highlighting denote the type of medal gain for each dataset, while underlined values indicate performance above the median. The symbol 'X' represents cases where methods failed to produce valid solutions within the allocated time constraints or encountered critical errors. As demonstrated in both subtables, our approach achieves superior performance on multiple datasets across various machine learning tasks, confirming the findings summarized in the main paper. These detailed results provide additional evidence of our method's robustness, versatility, and consistent performance advantages over existing techniques in automated machine learning systems.

## E.3 Detailed Error Analysis

To better understand the limitations of current AutoML systems, we conducted a comprehensive error analysis across MLZero, Codex CLI, AIDE, and DS-Agent (DS). Table 3 presents the frequency of distinct error types observed in the final iterations of each system's execution.

MLZero exhibits exceptional robustness, encountering minimal errors (6.5% overall) limited to algorithm implementation (4.3%) and preprocessing stages (2.1%). The algorithm implementation errors occurred specifically on sequence-to-sequence and audio classification tasks where the system had insufficient knowledge of relevant ML libraries. The preprocessing errors manifested exclusively on image-to-image tasks with heterogeneous input resolutions. In contrast, competing methods demonstrated substantially higher failure rates: DS-Agent (76.0%), AIDE (47.8%), and Codex CLI with reasoning (26.9%).

Table 18: Methods Performance on MLE-bench Lite. Gold/silver/bronze highlight the gold/silver/bronze medal, underline denotes above-median. Note that for metrics where lower values originally indicated better performance, we applied a negative sign to convert them, ensuring that higher values consistently represent better performance throughout the table. See table 6 for the dataset details.

| Method | D1 | D2 | D3 | D4 | D5 | D6 | D7 | D8 | D9 | D10 | D11 |
|---|---|---|---|---|---|---|---|---|---|---|---|
| Ours | 1.000 | 0.904 | X | 0.936 | -0.442 | -0.008 | 0.998 | 0.985 | -0.242 | X | -5.111 |
| AIDE | 1.000 | 0.855 | X | X | -0.694 | -0.817 | 0.996 | 0.903 | -0.801 | X | -5.463 |
| MLAB | 0.943 | 0.712 | X | 0.852 | -4.800 | -12.759 | X | 0.953 | X | X | -10.022 |
| OD | 0.495 | X | -0.220 | 0.884 | X | -0.426 | 0.853 | 0.971 | -0.934 | X | -1053.080 |

(a) Performance on Datasets from D1 to D11.

| Method | D12 | D13 | D14 | D15 | D16 | D17 | D18 | D19 | D20 | D21 |
|---|---|---|---|---|---|---|---|---|---|---|
| Ours | -0.059 | 0.990 | 0.787 | 0.673 | -0.384 | 0.963 | 0.960 | X | 0.958 | 0.625 |
| AIDE | -0.069 | 0.962 | 0.642 | 0.859 | -0.426 | 0.958 | 0.899 | 0.991 | X | 0.869 |
| MLAB | -0.063 | 0.817 | 0.500 | 0.421 | -0.555 | 0.943 | X | X | X | X |
| OD | -0.542 | 0.494 | 0.684 | 0.635 | -0.563 | 0.958 | X | X | X | 0.914 |

(b) Performance on Datasets from D12 to D22.

DS-Agent encounters difficulties at foundational stages, including perception (24.0%) and preprocessing (20.0%), indicating systemic failure in data comprehension and requirement translation, despite being provided with data perception context from MLZero. Its high algorithm implementation error rate (28.0%) likely stems from the substantial domain gap between deployment and development datasets.

AIDE's predominant failure mode is API hallucination (28.2%), wherein the system attempts to utilize non-existent or incorrectly implemented library functions. This underscores the necessity for external knowledge of ML libraries rather than exclusive reliance on parametric knowledge embedded in LLMs.

Notably, MLZero completely eliminates several error categories, including perception (0.0%), API hallucination (0.0%), and postprocessing (0.0%). This achievement can be attributed to our dual-memory architecture, which effectively anchors the LLM's generations in factually accurate external knowledge while maintaining execution consistency through structured episodic memory. The preprocessing errors observed in MLZero result from the system utilizing example file metadata to construct solutions for datasets with variable input resolutions across images, representing a conscious trade-off between context length compression and preservation of critical information in file grouping and perception.

It is important to note that this analysis focuses on final iteration errors and does not necessarily indicate error propensities at intermediate stages. For instance, AIDE's higher apparent API hallucination errors compared to DS-Agent may reflect AIDE's attempts at implementing more sophisticated solutions after successfully navigating earlier stages that DS-Agent fails to complete. Additionally, AIDE exhibits unreported errors (8.7%) in the MLE-Bench evaluation, potentially influencing comparative assessments.

These findings highlight the significant advantages of our integrated approach, which systematically addresses error-prone areas of the ML workflow through specialized perception agents, structured knowledge retrieval, and iterative refinement with targeted error correction mechanisms.

### E.4 Continual Improvement Beyond First Success

While the default MLZero configuration employs an early stopping strategy (terminating upon first successful execution), we investigated the system's capability for iterative refinement beyond initial success. We evaluated two configurations across eight challenging datasets covering all modalities: the default configuration with early stopping upon first success (def), and an extended

configuration allowing up to 5 iterations regardless of initial success (def + nostop). We also evaluated both configurations in the challenging setting without specialized external libraries (-ext) to assess improvement potential when the first solution is typically suboptimal.

Table 19 presents detailed results showing that the extended configuration demonstrates clear capability to refine successful solutions. Notable improvements include camoseg accuracy improving from 0.84 to 0.86 (+2.4%), flood from 0.69 to 0.74 (+7.2%), and solar RMSE dramatically reducing from 1.49 to 0.38 (-74.5%). The improvement pattern is more pronounced in the -ext setting where initial solutions are weaker due to lack of specialized libraries, with camoseg improving from 0.46 to 0.55 (+19.6%) and petfinder from 0.38 to 0.44 (+15.8%). These results validate that optimizing an existing working solution is generally easier than achieving the first success.

Table 19: Continual improvement results. Allowing iterations beyond first success improves performance but significantly increases computational cost. Metrics: higher is better for classification tasks; lower is better for regression tasks (marked with ↓). X indicates failure.

| Dataset | def | def + nostop | -ext | -ext + nostop |
|---|---|---|---|---|
| camoseg | 0.84 | 0.86 | 0.46 | 0.55 |
| flood | 0.69 | 0.74 | 0.60 | 0.62 |
| fiqa | 0.50 | 0.51 | 0.22 | 0.23 |
| mldoc | 0.95 | 0.96 | 0.94 | 0.96 |
| petfinder | 0.39 | 0.38 | 0.38 | 0.44 |
| rvlcdip | 0.87 | 0.87 | 0.89 | 0.88 |
| solar (10m)↓ | 1.49 | 0.38 | X | X |
| yolanda↓ | 8.53 | 8.54 | 8.93 | 8.75 |
| Avg Tokens↓ | 63k | 186k | 51k | 146k |
| Avg Time (s)↓ | 1904 | 5388 | 1731 | 4520 |

Despite performance gains, continual improvement incurs substantial computational overhead. Token usage increases $3\times$ from 63k to 186k tokens per dataset in the default setting, computation time nearly triples from 1904s to 5388s per dataset, and costs increase proportionally due to additional LLM calls. This overhead occurs because datasets that succeed in the first iteration still complete all 5 iterations rather than stopping early, continuing to explore alternative implementations even when initial performance is already strong.

Given these findings, we chose the early stopping strategy for the main paper to prioritize efficiency while maintaining high performance (92% success rate). The modest improvements from continual iteration (typically 2-7% gain) do not justify tripling computational costs for most practical applications. However, for scenarios where computational resources are not a constraint and optimal performance is paramount, the continual improvement capability remains valuable. As demonstrated in Table 1, extending the time limit to 24 hours with the `best_quality` preset further improves performance, approaching expert-level human results on several datasets.

## E.5 Examples of Generated Code

### E.5.1 Code generated by MLZero def on the Hateful Meme Dataset

Evaluation Score (f1): 0.5868

Time Used: 663 seconds

```
"""                                                                          1
Hateful Meme Classification using AutoGluon MultiModal                        2
                                                                              3
This script uses AutoGluon's MultiModalPredictor to classify memes as         4
    hateful (1) or not hateful (0)
based on both image content and text. The model processes image-text          5
    pairs to make binary predictions.
                                                                              6
Installation requirements:                                                    7
pip install autogluon.multimodal                                              8
```

```
pip install pandas                                                          9
pip install numpy                                                          10
"""                                                                        11
                                                                           12
import os                                                                  13
import pandas as pd                                                        14
import numpy as np                                                         15
import datetime                                                            16
from autogluon.multimodal import MultiModalPredictor                       17
                                                                           18
def expand_image_paths(df, base_path):                                     19
    """                                                                    20
    Expands relative image paths to absolute paths                         21
                                                                           22
    Args:                                                                  23
        df: DataFrame containing image paths                               24
        base_path: Base directory for images                               25
                                                                           26
    Returns:                                                               27
        DataFrame with expanded image paths                                28
    """                                                                    29
    df = df.copy()                                                         30
    # Convert image paths to absolute paths if they are not already        31
    if 'img' in df.columns:                                                32
        df['img'] = df['img'].apply(lambda x: os.path.join(base_path,      33
    x) if not os.path.isabs(x) else x)
        return df                                                          34
                                                                           35
def preprocess_data(train_df, test_df):                                    36
    """                                                                    37
    Preprocesses training and test data                                    38
                                                                           39
    Args:                                                                  40
        train_df: Training DataFrame                                       41
        test_df: Test DataFrame                                            42
                                                                           43
    Returns:                                                               44
        Preprocessed training and test DataFrames                          45
    """                                                                    46
    # Remove unnecessary index column if present                          47
    if 'Unnamed: 0' in train_df.columns:                                   48
        train_df = train_df.drop(columns=['Unnamed: 0'])                   49
    if 'Unnamed: 0' in test_df.columns:                                    50
        test_df = test_df.drop(columns=['Unnamed: 0'])                     51
                                                                           52
    # Drop rows with NA values in training data                           53
    train_df = train_df.dropna(subset=['label'])                          54
                                                                           55
    return train_df, test_df                                              56
                                                                           57
def train_model(train_data, model_save_path):                             58
    """                                                                    59
    Trains a MultiModalPredictor model                                     60
                                                                           61
    Args:                                                                  62
        train_data: Training data DataFrame                                63
        model_save_path: Path to save the trained model                    64
                                                                           65
    Returns:                                                               66
        Trained MultiModalPredictor                                        67
    """                                                                    68
    # Initialize and train the model                                      69
    predictor = MultiModalPredictor(                                      70
        label='label',                                                    71
        path=model_save_path,                                             72
```

```python
        problem_type='binary',   # Binary classification task        73
    )                                                                 74
                                                                      75
    # Train the model with medium quality preset and time limit       76
    predictor.fit(                                                    77
        train_data=train_data,                                        78
        time_limit=1800,   # 30 minutes                               79
        presets="medium_quality"                                      80
    )                                                                 81
                                                                      82
    return predictor                                                  83
                                                                      84
def make_predictions(predictor, test_data, output_path):             85
    """                                                               86
    Makes predictions on test data and saves results                 87
                                                                      88
    Args:                                                             89
        predictor: Trained MultiModalPredictor                        90
        test_data: Test data DataFrame                                91
        output_path: Path to save predictions                         92
    """                                                               93
    # Generate predictions                                            94
    predictions = predictor.predict(test_data)                       95
                                                                      96
    # Create results DataFrame with the same format as test data      97
    results = test_data.copy()                                        98
    results['label'] = predictions                                   99
                                                                     100
    # Save predictions to CSV                                        101
    results_path = os.path.join(output_path, "results.csv")         102
    results.to_csv(results_path, index=False)                        103
                                                                     104
    print(f"Predictions saved to {results_path}")                   105
                                                                     106
if __name__ == "__main__":                                          107
    # Define paths                                                   108
    output_dir = "/media/agent/maab/runs/RUN_20250504_222338/outputs/   109
    agentsonnet37_hateful_meme_output"
    os.makedirs(output_dir, exist_ok=True)                          110
                                                                     111
    # Create a timestamped folder for the model                     112
    timestamp = datetime.datetime.now().strftime("%Y%m%d_%H%M%S")   113
    model_dir = os.path.join(output_dir, f"model_{timestamp}")      114
    os.makedirs(model_dir, exist_ok=True)                           115
                                                                     116
    # Load data                                                      117
    train_data_path = "/media/agent/maab/datasets/hateful_meme/      118
    training/train.csv"
    test_data_path = "/media/agent/maab/datasets/hateful_meme/training   119
    /inference.csv"
    image_base_path = "/media/agent/maab/datasets/hateful_meme/      120
    training"
                                                                     121
    train_df = pd.read_csv(train_data_path)                         122
    test_df = pd.read_csv(test_data_path)                           123
                                                                     124
    # Preprocess data                                               125
    train_df, test_df = preprocess_data(train_df, test_df)         126
                                                                     127
    # Expand image paths                                            128
    train_df = expand_image_paths(train_df, image_base_path)       129
    test_df = expand_image_paths(test_df, image_base_path)         130
                                                                     131
    print(f"Training data shape: {train_df.shape}")                132
    print(f"Test data shape: {test_df.shape}")                     133
```

```
                                                                                    134
    # Train model                                                                   135
    predictor = train_model(train_df, model_dir)                                    136
                                                                                    137
    # Make predictions                                                              138
    make_predictions(predictor, test_df, output_dir)                               139
                                                                                    140
    print("Hateful meme classification completed successfully!")                    141
```

### E.5.2 Code generated by Codex +rea on the Hateful Meme Dataset

Evaluation Score (f1): 0.4538

Time Used: 110 seconds

```python
#!/usr/bin/env python3                                                              1
"""                                                                                 2
Baseline script for Hateful Meme classification using TF-IDF and                   3
    Logistic Regression.
Loads training data, trains a text-only model, and predicts on                     4
    inference data.
Saves predictions to 'results.csv' in the current directory.                        5
"""                                                                                 6
import os                                                                            7
import pandas as pd                                                                  8
                                                                                    9
from sklearn.feature_extraction.text import TfidfVectorizer                         10
from sklearn.linear_model import LogisticRegression                                11
                                                                                    12
def main():                                                                         13
    # Paths to data                                                                 14
    train_path = "/media/agent/maab/datasets/hateful_meme/training/                 15
    train.csv"
    test_path = "/media/agent/maab/datasets/hateful_meme/training/                  16
    inference.csv"
    # Load datasets                                                                 17
    train_df = pd.read_csv(train_path)                                             18
    test_df = pd.read_csv(test_path)                                               19
                                                                                    20
    # Text vectorization                                                            21
    vectorizer = TfidfVectorizer(max_features=5000, ngram_range=(1, 2)             22
    , stop_words='english')
    X_train = vectorizer.fit_transform(train_df['text'].astype(str))              23
    y_train = train_df['label']                                                    24
                                                                                    25
    # Train classifier                                                              26
    clf = LogisticRegression(solver='liblinear', max_iter=1000)                    27
    clf.fit(X_train, y_train)                                                       28
                                                                                    29
    # Predict on test data                                                          30
    X_test = vectorizer.transform(test_df['text'].astype(str))                    31
    preds = clf.predict(X_test)                                                     32
                                                                                    33
    # Prepare and save results                                                      34
    _, ext = os.path.splitext(test_path)                                           35
    output_path = os.path.join(os.getcwd(), f"results{ext}")                       36
    result_df = pd.DataFrame({                                                      37
        'id': test_df['id'],                                                        38
        'label': preds                                                              39
    })                                                                              40
    result_df.to_csv(output_path, index=False)                                     41
    print(f"Saved results to {output_path}")                                       42
                                                                                    43
if __name__ == '__main__':                                                          44
    main()                                                                          45
```

### E.5.3 Code generated by AIDE def on the Hateful Meme Dataset

Evaluation Score (f1): 0.5573

Time Used: 2672 seconds

```python
import os
import pandas as pd
import numpy as np
import torch
import torch.nn as nn
import torch.nn.functional as F
from torch.utils.data import Dataset, DataLoader
from torchvision import transforms, models
from transformers import DistilBertTokenizer, DistilBertModel
from PIL import Image
from sklearn.model_selection import StratifiedKFold
from sklearn.metrics import accuracy_score, f1_score
import warnings

warnings.filterwarnings("ignore")

# Set device
device = torch.device("cuda" if torch.cuda.is_available() else "cpu")
print(f"Using device: {device}")

# Load data
train_df = pd.read_csv("./input/train.csv")
test_df = pd.read_csv("./input/inference.csv")

# Check label distribution
print(f"Label distribution: {train_df['label'].value_counts()}")

# Define image transforms
image_transforms = transforms.Compose(
    [
        transforms.Resize((224, 224)),
        transforms.ToTensor(),
        transforms.Normalize(mean=[0.485, 0.456, 0.406], std=[0.229,
    0.224, 0.225]),
    ]
)

# Initialize tokenizer
tokenizer = DistilBertTokenizer.from_pretrained("distilbert-base-
    uncased")

# Define dataset class
class MultimodalDataset(Dataset):
    def __init__(self, dataframe, transform=None, is_test=False):
        self.dataframe = dataframe
        self.transform = transform
        self.is_test = is_test

    def __len__(self):
        return len(self.dataframe)

    def __getitem__(self, idx):
        row = self.dataframe.iloc[idx]

        # Load and transform image
        img_path = os.path.join("./input", row["img"])
        try:
            image = Image.open(img_path).convert("RGB")
            if self.transform:
```

```
                image = self.transform(image)                          59
        except Exception as e:                                          60
            print(f"Error loading image {img_path}: {e}")               61
            # Create a blank image if there's an error                  62
            image = torch.zeros(3, 224, 224)                            63
                                                                        64
        # Tokenize text                                                 65
        text = str(row["text"])                                         66
        encoding = tokenizer(                                           67
            text,                                                       68
            return_tensors="pt",                                        69
            padding="max_length",                                       70
            truncation=True,                                            71
            max_length=128,                                             72
        )                                                               73
                                                                        74
        # Get input_ids and attention_mask                             75
        input_ids = encoding["input_ids"].squeeze(0)                    76
        attention_mask = encoding["attention_mask"].squeeze(0)          77
                                                                        78
        if self.is_test:                                                79
            return {                                                    80
                "id": row["id"],                                        81
                "image": image,                                         82
                "input_ids": input_ids,                                 83
                "attention_mask": attention_mask,                       84
            }                                                           85
        else:                                                           86
            return {                                                    87
                "id": row["id"],                                        88
                "image": image,                                         89
                "input_ids": input_ids,                                 90
                "attention_mask": attention_mask,                       91
                "label": torch.tensor(row["label"], dtype=torch.long),  92
            }                                                           93
                                                                        94
                                                                        95
# Define model                                                          96
class MultimodalModel(nn.Module):                                       97
    def __init__(self):                                                 98
        super(MultimodalModel, self).__init__()                         99
                                                                        100
        # Image encoder (ResNet50)                                      101
        self.image_encoder = models.resnet50(pretrained=True)           102
        for param in self.image_encoder.parameters():                  103
            param.requires_grad = False                                 104
        self.image_encoder.fc = nn.Linear(self.image_encoder.fc.        105
    in_features, 512)
                                                                        106
        # Text encoder (DistilBERT)                                     107
        self.text_encoder = DistilBertModel.from_pretrained("           108
    distilbert-base-uncased")
        for param in self.text_encoder.parameters():                  109
            param.requires_grad = False                                 110
        self.text_pooler = nn.Linear(768, 512)                          111
                                                                        112
        # Classifier                                                    113
        self.classifier = nn.Sequential(                                114
            nn.Linear(1024, 512),                                       115
            nn.ReLU(),                                                  116
            nn.Dropout(0.3),                                            117
            nn.Linear(512, 256),                                        118
            nn.ReLU(),                                                  119
            nn.Dropout(0.3),                                            120
            nn.Linear(256, 2),                                          121
```

```python
        )

    def forward(self, image, input_ids, attention_mask):
        # Image features
        image_features = self.image_encoder(image)

        # Text features
        text_output = self.text_encoder(
            input_ids=input_ids, attention_mask=attention_mask
        )
        text_features = text_output.last_hidden_state[:, 0, :]  # Use
    [CLS] token
        text_features = self.text_pooler(text_features)

        # Concatenate features
        combined_features = torch.cat((image_features, text_features),
     dim=1)

        # Classification
        output = self.classifier(combined_features)
        return output

# Training function
def train_model(model, train_loader, optimizer, criterion, device):
    model.train()
    total_loss = 0

    for batch in train_loader:
        image = batch["image"].to(device)
        input_ids = batch["input_ids"].to(device)
        attention_mask = batch["attention_mask"].to(device)
        labels = batch["label"].to(device)

        optimizer.zero_grad()
        outputs = model(image, input_ids, attention_mask)
        loss = criterion(outputs, labels)
        loss.backward()
        optimizer.step()

        total_loss += loss.item()

    return total_loss / len(train_loader)

# Evaluation function
def evaluate_model(model, val_loader, criterion, device):
    model.eval()
    total_loss = 0
    all_preds = []
    all_labels = []

    with torch.no_grad():
        for batch in val_loader:
            image = batch["image"].to(device)
            input_ids = batch["input_ids"].to(device)
            attention_mask = batch["attention_mask"].to(device)
            labels = batch["label"].to(device)

            outputs = model(image, input_ids, attention_mask)
            loss = criterion(outputs, labels)

            total_loss += loss.item()

            _, preds = torch.max(outputs, 1)
```

```python
                all_preds.extend(preds.cpu().numpy())                          185
                all_labels.extend(labels.cpu().numpy())                        186

        accuracy = accuracy_score(all_labels, all_preds)                       188
        f1 = f1_score(all_labels, all_preds)                                   189

        return total_loss / len(val_loader), accuracy, f1                      191

# Prediction function                                                          194
def predict(model, test_loader, device):                                       195
    model.eval()                                                               196
    predictions = []                                                           197
    ids = []                                                                   198

    with torch.no_grad():                                                      200
        for batch in test_loader:                                             201
            image = batch["image"].to(device)                                 202
            input_ids = batch["input_ids"].to(device)                         203
            attention_mask = batch["attention_mask"].to(device)              204
            batch_ids = batch["id"]                                           205

            outputs = model(image, input_ids, attention_mask)                207
            _, preds = torch.max(outputs, 1)                                  208

            predictions.extend(preds.cpu().numpy())                          210
            ids.extend(batch_ids.numpy())                                    211

    return ids, predictions                                                   213

# Cross-validation setup                                                       216
n_splits = 5                                                                   217
skf = StratifiedKFold(n_splits=n_splits, shuffle=True, random_state           218
    =42)

# Hyperparameters                                                              220
batch_size = 16                                                                221
num_epochs = 3                                                                 222
learning_rate = 1e-4                                                           223

# Lists to store metrics                                                       225
cv_accuracies = []                                                             226
cv_f1_scores = []                                                              227

# For storing test predictions from each fold                                 229
test_predictions = []                                                          230

# Create test dataset and loader                                              232
test_dataset = MultimodalDataset(test_df, transform=image_transforms,        233
    is_test=True)
test_loader = DataLoader(test_dataset, batch_size=batch_size, shuffle=        234
    False)

# Cross-validation loop                                                        236
for fold, (train_idx, val_idx) in enumerate(skf.split(train_df,              237
    train_df["label"])):
    print(f"\nFold {fold + 1}/{n_splits}")                                    238

    # Split data                                                              240
    train_data = train_df.iloc[train_idx].reset_index(drop=True)             241
    val_data = train_df.iloc[val_idx].reset_index(drop=True)                 242

    # Create datasets and dataloaders                                        244
```

```
    train_dataset = MultimodalDataset(train_data, transform=      245
image_transforms)
    val_dataset = MultimodalDataset(val_data, transform=          246
image_transforms)
                                                                  247
    train_loader = DataLoader(train_dataset, batch_size=batch_size, 248
shuffle=True)
    val_loader = DataLoader(val_dataset, batch_size=batch_size,   249
shuffle=False)
                                                                  250
    # Initialize model, optimizer, and loss function            251
    model = MultimodalModel().to(device)                         252
    optimizer = torch.optim.Adam(model.parameters(), lr=learning_rate) 253
    criterion = nn.CrossEntropyLoss()                            254
                                                                  255
    # Training loop                                              256
    for epoch in range(num_epochs):                             257
        train_loss = train_model(model, train_loader, optimizer, 258
criterion, device)
        val_loss, accuracy, f1 = evaluate_model(model, val_loader, 259
criterion, device)
                                                                  260
        print(                                                   261
            f"Epoch {epoch+1}/{num_epochs}, Train Loss: {train_loss:.4 262
f}, Val Loss: {val_loss:.4f}, Accuracy: {accuracy:.4f}, F1: {f1:.4
f}"
        )                                                         263
                                                                  264
    # Final evaluation                                           265
    _, accuracy, f1 = evaluate_model(model, val_loader, criterion, 266
device)
    cv_accuracies.append(accuracy)                               267
    cv_f1_scores.append(f1)                                      268
                                                                  269
    print(f"Fold {fold + 1} - Final Accuracy: {accuracy:.4f}, F1: {f1 270
:.4f}")
                                                                  271
    # Make predictions on test set                              272
    ids, preds = predict(model, test_loader, device)            273
    test_predictions.append(preds)                              274
                                                                  275
# Calculate average metrics                                     276
mean_accuracy = np.mean(cv_accuracies)                          277
mean_f1 = np.mean(cv_f1_scores)                                 278
std_accuracy = np.std(cv_accuracies)                           279
std_f1 = np.std(cv_f1_scores)                                  280
                                                                  281
print(f"\nCross-Validation Results:")                          282
print(f"Mean Accuracy: {mean_accuracy:.4f} +- {std_accuracy:.4f}") 283
print(f"Mean F1 Score: {mean_f1:.4f} +- {std_f1:.4f}")         284
                                                                  285
# Average test predictions from all folds                       286
test_predictions = np.array(test_predictions)                   287
final_predictions = np.round(np.mean(test_predictions, axis=0)).astype 288
    (int)
                                                                  289
# Create submission file                                        290
submission = pd.DataFrame({"id": test_df["id"], "label":        291
    final_predictions})
                                                                  292
# Save submission file                                          293
submission.to_csv("./working/submission.csv", index=False)      294
print(f"Submission file saved to ./working/submission.csv")     295
```

# F    More Related Work: AutoML with LLMs

Recent advances have demonstrated the potential of integrating LLMs into AutoML systems across various components of the ML workflow. For feature engineering, researchers [29, 82] have leveraged LLMs to generate meaningful features with explanations and efficiently evaluate candidate features. In hyperparameter optimization, LLM-based approaches [79, 47] generate configurations through dataset analysis and iterative refinement processes. For neural architecture search, researchers have combined LLMs with Quality-Diversity algorithms to generate diverse network architectures [55] and constructed performance predictors for estimating neural network efficiency [35]. LLMs have also shown effectiveness for data preprocessing tasks such as error detection and imputation [78]. While these approaches successfully apply LLMs to specific components, MLZero differs by providing an end-to-end multi-agent solution that fully automates the entire ML workflow.

# G    Limitations and Future Work

Despite MLZero's strong performance across diverse machine learning tasks, several limitations warrant consideration. First, while our system can leverage both documentation and code examples, its effectiveness may be limited when working with ML libraries that have minimal description in either text or code format. Second, machine learning libraries contain unavoidable undocumented bugs that are difficult to address through in-context learning, and LLMs cannot anticipate these issues unless explicitly prompted. Additionally, our experiments reveal that smaller LLMs (8B parameters) still demonstrate a significant performance gap compared to larger models despite showing competitive results.

Future research directions should address these challenges through several avenues. First, enhancing performance with smaller language models through improved alignment with external knowledge and curated code examples represents a promising direction, especially considering our 8B parameter model already demonstrates reasonable performance. Second, developing a finetuning process that allows LLMs to actively experiment with and learn ML libraries during training could help address both undocumented bugs and improve smaller model performance, while also expanding MLZero's applicability to emerging libraries with less structured documentation. This approach would enable models to acquire practical knowledge about the ML library behavior beyond what is explicitly documented. Finally, conducting longitudinal studies on real-world ML workflows would provide insights for better supporting human-AI collaboration in data science processes, particularly for iterative development scenarios where expert guidance can further improve automated approaches.

