# OpenReview forum: "MLZero: A Multi-Agent System for End-to-end Machine Learning Automation"
_NeurIPS.cc/2025/Conference — NeurIPS 2025 poster_

### Official Review · Reviewer_b88G · 2025-06-04

**Clarity:** 3
**Significance:** 3
**Originality:** 2
**Rating:** 4
**Confidence:** 4

**Summary:**

This paper presents MLZero, a hierarchical multi-agent system for fully automated machine learning (AutoML) across multimodal data. The framework integrates perception agents, semantic memory (knowledge retrieval), episodic memory (error-driven refinement), and iterative code generation based on LLMs, aiming for zero human intervention. MLZero is benchmarked on both MLE-Bench Lite and a new Multimodal AutoML Agent Benchmark introduced by the authors, showing competitive results in success rate and performance against various baselines. Extensive ablation and error analyses are also provided to justify architectural design choices.

**Questions:**

1. **Benchmark Diversity**: How were the 25 datasets in your new benchmark chosen? Is there any risk that the benchmark favors MLZero due to design alignment?
2. **Failure Analysis Detail**: For the minimal 6.5% error rate in MLZero, what are the qualitative patterns of failure? Are these reproducible errors or stochastic?
3. **Knowledge Contamination Risk**: Given that some agents access condensed tutorials and summaries, how do you ensure no contamination from evaluation sets?
4. **Memory Scalability**: How does episodic memory scale with longer runs or larger datasets? Is there a cost trade-off in performance vs. token consumption that might make it infeasible in practice?
5. **Security and Robustness**: Does MLZero ever produce unsafe or incorrect solutions that *look* correct (e.g., hallucinated accuracy but failed logic)? Any defenses?

*A more favorable evaluation could be reached if the authors:*

* Provide clearer guarantees or at least empirical error bounds for LLM planning and execution cycles.
* Include human validation of at least a subset of generated pipelines.
* Address benchmark neutrality through external evaluation or open access submission.

**Ethical Concerns:**

["NO or VERY MINOR ethics concerns only"]

**Final Justification:**

I have read the author response and appreciate the detailed clarifications and additional experiments. My main concerns — including benchmark fairness, system novelty, and error analysis — have been addressed satisfactorily. The paper is a solid empirical systems contribution with thoughtful design and extensive evaluation.

I am therefore increasing my score to 4.

**Limitations:**

The authors acknowledge runtime cost and large-model reliance, but do not adequately address:

* **Generalization to new ML libraries**: It is unclear how easily MLZero adapts to unseen or rapidly changing APIs.
* **Dependence on LLM correctness**: The system heavily assumes the base LLM will not hallucinate or misinterpret errors in a critical way.
* **Memory curation and freshness**: Episodic and semantic memories appear stateless across datasets—how would they behave in a real, long-term deployed setting?

**Quality:**

4

**Strengths And Weaknesses:**

#### **Strengths**

* **Well-structured System Design**: The paper presents a clear and modular architecture, combining LLM planning and execution with memory augmentation and agent specialization.
* **Multimodal Generality**: MLZero operates on diverse tasks spanning tabular, text, image, and document classification, which showcases practical versatility.
* **Comprehensive Evaluation**: Experiments span two benchmarks, including a new 25-task suite with raw data processing requirements.
* **Ablation and Failure Analysis**: The inclusion of ablation studies and error breakdown by category (e.g., hallucinations, preprocessing) is commendable and unusually thorough.
* **Competitive Results**: MLZero outperforms existing baselines in success rate and average rank, even when stripped of certain components like external knowledge or episodic memory.

#### **Weaknesses**

1. **Excessive Engineering Emphasis, Weak Novelty in Learning**: While the architecture is elaborate, most techniques (e.g., retrieval, summarization, error tracing) are standard LLM engineering practices. There is limited novelty from a learning or algorithmic standpoint.
2. **Limited Insight into LLM Limitations**: The reliance on LLMs is central, but the paper does not examine when/how LLMs fail during ML pipeline construction, or what qualitative reasoning errors occur.
3. **Benchmark Bias**: The newly introduced benchmark is entirely authored by the same authors. While it addresses limitations of MLE-Bench, the fairness and diversity of its design should be externally validated.
4. **Lack of Human Evaluation or Expert Annotation**: The paper focuses solely on automatic metrics, without any human judgment of code quality, correctness, or failure modes—this makes the empirical superiority somewhat opaque.
5. **Minimal Theoretical or Formal Analysis**: There is no formal grounding in agent planning, LLM reliability, or bounded correctness. A system this complex could benefit from guarantees or at least probabilistic behavior analysis.
6. **Missing Comparison with Planning-based LLM Approaches**: Other recent planning-agent frameworks (e.g., ReAct, Reflexion, CAMEL) are not directly compared or discussed in depth. This weakens claims of generality and superiority.

---

> ### Author Rebuttal · Authors · 2025-07-29
>
> To address 17 bulletpoints (weakness: 6 bulletpoints, questions: 5+3=8 bulletpoints, and limitations sections: 3 bulletpoints) comprehensively and concisely, we've organized our response into three categories: Methodology, Benchmark Design, and Evaluation.
>
> ---
>
> # Methodology (Q1, 10, 16, 17)
>
> ## Q1: Excessive Engineering Emphasis, Weak Novelty in Learning
>
> We respectfully disagree with the characterization of our work as "standard LLM engineering." While we use established techniques, our key contribution is the new and cohesive framework with 9 agents coordinating synergetically for end-to-end AutoML on raw, multimodal data. The synergistic combination of four modules and nine specialized agents creates a system that empirically eliminates entire classes of errors and achieves significant performance gains over existing approaches. Our ablation studies demonstrate that **simply adding semantic memory modules to other methods or using alternative approaches results in substantially degraded performance, highlighting that the specific integration and coordination of these components is essential.** This is fundamentally an empirical systems paper where the novel architecture and its rigorous validation constitute the core scientific contribution **[e.g. MLAgentBench (ICML'24), DS-Agent (ICML'24), MDAgents (Neurips'24 Oral), etc.]**.
>
> ## Q10: Memory Scalability
>
> First, episodic memory consumption is independent of dataset size - it only depends on the number of failed steps. Our system adds approximately 150 tokens per failed step to the context through our Error Analyzer agent's efficient condensation, and can theoretically support more than 100 steps while staying within context limits. As for the cost trade-off question, while continuing iterations beyond first success can improve performance (up to +0.09 accuracy gain, as detailed in our response to Q3 to reviewer aYBU), the initial successful solution already achieves strong results. The memory overhead remains manageable since most tasks succeed within 3 steps, making the system practical and cost-effective. Users can always trade additional computational resources for incremental improvements and our system is scalable to support that.
>
> ## Q16: Dependence on LLM correctness
>
> On the contrary, we have specifically engineered our system to **address LLM limitations through our multi-agent architectural design**. Our system is deliberately structured to detect and recover from LLM failures through multi-agent synergy and iterative refinement. When one agent makes an error, others can detect and correct it, preventing fatal failures. Our experimental results confirm this robustness—across all experiments, we encountered no failures due to early-stage perception errors, and our stress tests with deliberately misleading descriptions ( as detailed in our response to Q4 to reviewer dDLh) still resulted in correct task perception. Furthermore, Tab 1 shows that even without external knowledge or episodic memory components, or using a very small 8B LLM, performance remains superior to competitors.
>
> ## Q17: Memory curation and freshness
>
> **For fair benchmarking, our system runs independently on each task**, with memories reset between tasks to prevent information leakage. In a real-world deployment, both memory systems would provide additional benefits: the semantic memory can be continuously updated with new library documentation, best practices, and community knowledge; the episodic memory would accumulate valuable insights across similar tasks, improving efficiency through transfer learning. The modular design allows for easy updates to both memory systems, ensuring continued relevance and performance as ML tools and techniques evolve.
>
> ---
>
> # Benchmark Design (Q3, 7, 9, 11, 14)
>
> ## Q3, Q7, Q14: Benchmark Bias, Diversity, and Neutrality
>
> **We clarify that MLZero achieves superior results on the existing MLE-Bench Lite benchmark before introducing our own benchmark, which was designed independently without bias towards MLZero.** We created the Multimodal AutoML Agent Benchmark (MAAB) out of necessity, as existing benchmarks don't support evaluation on raw, unprocessed data—a critical gap that masks real-world automation challenges.
>
> To ensure fairness and diversity, MAAB is built on 25 diverse public datasets from reputable sources (Kaggle, UCI, BEIR) selected from machine learning publications, **most without direct open-source solutions**. It also spans 8 distinct problem types (classification, segmentation, retrieval, etc.) across multiple modalities (tabular, text, image, document data).
>
> We will open-source the entire benchmark, including all datasets and evaluation code, for external validation and community use (included in supplemental materials).
>
> ## Q9: Knowledge Contamination Risk
>
> Our knowledge base contains only public, official tutorial documentation for ML libraries themselves—no information about evaluation datasets or their solutions. The dataset does not appear in our knowledge base, and processing raw data is fundamentally different from using pre-processed tutorial examples. More importantly, our evaluation datasets and their test sets are completely held out. This follows standard RAG practices [Retrieval-Augmented Generation for Large Language Models: A Survey] where external knowledge remains general and task-agnostic.
>
> ## Q11: Security and Robustness
>
> **During benchmarking, test data is completely held out from all agents, eliminating any possibility of hallucinated accuracy claims**. Our system itself also includes robust safeguards against producing incorrect solutions that appear correct. The Executer agent doesn't simply check for zero exit codes but uses an LLM to analyze execution logs and outputs to determine logical success. It can trigger a "FIX" decision even when code runs without crashing if it detects logical errors or poor performance (e.g., accuracy of 0.01), ensuring higher solution reliability.
>
> ---
>
> # Evaluations (Q2, 4, 5, 6, 8, 12, 13, 15)
>
> ## Q2: Limited Insight into LLM Limitations
>
> We respectfully disagree with this point. Our paper provides a very detailed analysis of LLM limitations in this context **in Section 4.4 (Failure Cases and Analysis) and Appendix E.3 (Detailed Error Analysis)**. Due to context limitation, we'll not repeat here.
>
> ## Q4, Q13: Lack of Human Evaluation/Validation for code quality, correctness, or failure modes
>
> Note that we prioritized automated metrics for task success (**correctness**) because they are objective, scalable, and directly measure the system's ability to solve the problem. Given that **code quality** is important, we have included full, verbatim examples of generated code from MLZero, Codex CLI, and AIDE in **Appendix E.4**. These examples allow for a direct qualitative comparison. For instance, the MLZero-generated code for the Hateful Meme task (Appendix E.4.1) is modular, well-commented, and uses a sophisticated multimodal approach, contrasting with the simpler, text-only logistic regression from Codex CLI (E.4.2) and the more complex but less efficient boilerplate from AIDE (E.4.3). **As stated above in Q2, we have included expert annotation of failure modes in main paper and appendix.**
>
> ## Q5, Q6: Minimal Theoretical Analysis & Comparison with Planning-based LLM Approaches
>
> Our work is an empirical contribution focused on building and validating a complex system. Formal guarantees for LLM-based agentic systems are a nascent and very challenging open research problem beyond the scope of this paper.
>
> Regarding planning agents like ReAct, Reflexion, or CAMEL, these are general reasoning frameworks, not end-to-end ML Agent systems, therefore can't be compared directly. Furthermore, we do include **Codex CLI—a state-of-the-art planning agent approach**—as a baseline in Table 1. However, per the reviewer's request, we tested CAMEL on our benchmark because it is still actively maintained among those three methods for the updated API calls. The results below show CAMEL struggled significantly, with most attempts failing (marked with "X") across three runs (12.5% overall success rate):
>
> | CAMEL setting: GPT-4o, with CodeExecutionToolkit and FileWriteToolkit | Run 1 | Run 2 | Run 3 |
> |:---------------------------------------------------------------------:|:-----:|:-----:|:-------:|
> | camoseg | X | X | X |
> | flood | X | X | X |
> | fiqa | X | X | X |
> | mldoc | X | X | 0.63|
> | petfinder | X | X | 0.34 |
> | rvlcdip | X | X | X |
> | solar(10m)↓ | X | X | X |
> | yolanda↓ | X | 9.62 | X |
>
> ## Q8, Q12: Failure Analysis Detail & Empirical Error Bounds
>
> Our analysis of MLZero's 6.5% error rate shows non-deterministic (Table 6) failures that vary between LLM runs, stemming from two main sources: algorithm implementation errors (\~4.3%) on tasks lacking specialized libraries (particularly audio processing) and preprocessing errors (\~2.1%) on image-to-image tasks with heterogeneous input sizes.
>
> Despite this variability in specific implementations, our manual annotation of 46 datasets provides statistical confidence in the reported **6.5% error rate (standard deviation \~3.6%)**, while establishing empirical reliability bounds with a **92% success rate (from 75 runs, standard deviation \~3.1%)** across diverse tasks. Although formal guarantees for LLM systems remain an open research challenge, our systematic error categorization and ablation studies offer valuable insights into failure modes and component contributions to reliability. This comprehensive empirical framework demonstrates that while individual errors may differ between runs due to the probabilistic nature of LLM outputs, the overall error patterns remain consistent enough to inform targeted improvements in future work.
>
> ## Q15: Generalization to new ML libraries
>
> **One of our key contributions is a system designed for easy integration of new ML libraries.** Due to context limitation, please refer to the answer to Q2 of reviewer dDLh.

---

> > ### Comment · Reviewer_b88G · 2025-08-01
> >
> > Thank you for the detailed and well-organized response. The authors have addressed my concerns clearly, especially regarding the benchmark construction, system novelty, and failure analysis. I have updated my score accordingly.

---

> > > ### Author Response · Authors · 2025-08-01
> > >
> > > Thank you for your detailed review and for taking the time to consider our rebuttal. We appreciate your careful attention to our responses. Your feedback has been valuable to improving our work.

---

### Official Review · Reviewer_aYBU · 2025-06-24

**Clarity:** 4
**Significance:** 2
**Originality:** 3
**Rating:** 4
**Confidence:** 4

**Summary:**

The paper proposes a multi-agent system titled “MLZero” which achieves impressive results on MLE Bench and their own Multimodal AutoML benchmark while being fully automated. The paper discussed their system setup and agent definitions along with the benchmarks.

**Questions:**

1. There is no mention of which LLM is used in the “def” setting of MLZero. Could this be added to the paper? The config present in the appendix section seems to indicate this is Claude 3.7 Sonnet, but it would be great if the authors could confirm this.
2. The agent seems to “finish execution” once it successfully executes the code (or reaches the context limit). It would’ve been great to see the agent continually create better solutions and maintain a track of the best solution. Ablations to see the improvements from this setup versus the current would’ve provided valuable insights into the self improvement capabilities of LLMs (something like AlphaEvolve). I’m not sure if these ablations were done. Could you add details about those if they were done?

**Ethical Concerns:**

["NO or VERY MINOR ethics concerns only"]

**Limitations:**

Yes

**Quality:**

4

**Strengths And Weaknesses:**

Strengths:
1. The paper is very detailed and well written.
2. They achieve impressive results on the MLE Bench and Multimodal AutoML benchmarks.
3. The paper contains extensive ablations and detailed task-level results on the various ML benchmarks.

Weaknesses:
1. While the results are impressive, the system itself is simple and doesn't represent a significant contribution to the field. The crux of the paper lies in the setup of the system and the way the different agents interact with each other, and while that is discussed in fair detail, the system isn’t complex or novel enough to meaningfully move the field forward.

---

> ### Author Rebuttal · Authors · 2025-07-29
>
> We thank you for your feedback and are glad you found the paper well-written and the results impressive.
>
> ---
>
> ## Q1: System Simplicity and Contribution
>
> Our innovation lies in how we invent **a new synergetic multi-agent system for the AutoML domain, which is challenging given the total number of the agents (9 in total) in MLZero**. While the individual components are inspired by previous literature under different contexts. The system innovation can be essential **while other alternative approach greatly deteriorate the performance** (see ablation study in **Section 4.2.2 and 4.3**). Similarly, adding memory module to other methods (e. g. AIDE) still can't outperform MLZero (Table 1). Those evidence show that the synergy can be essential for AutoML applications. Furthermore, The effectiveness of this specific synergy is demonstrated by our complete elimination of API hallucination and perception errors (Table 3), which are primary failure modes for strong baselines like AIDE and DS-Agent. Its significance lies in achieving effectiveness (+263.6% improvement) and state-of-the-art performance in end-to-end ML automation. **The novelty has been mentioned in L55.**
>
> Because using LLM agent to propose high quality ML solution are considered to be very challenging and requires **higher entry bar [MLAgentBench (ICML'24)]**, hence MLZero can be potentially adaptable to broader data science related tasks with minimal modifications, which can inspire future work in this community.
>
> ---
>
> ## Q2: Which LLM is used in the "def" setting?
>
> The LLM used in the def configuration is **Claude 3.7 Sonnet**. This is specified in the implementation details in **Appendix C.1** and the configuration file snippet in **Appendix C.2**.
>
> ---
>
> ## Q3: Continual Improvement of Solutions
>
> **Without the early stopping, MLZero can continue to improve the performance after first success** (up to +0.09 accuracy), we have conducted experiments exploring continuous improvement across 8 ablation datasets as shown in the table below:
>
> | Dataset | def | def + nostop | -ext | -ext + nostop |
> |:------------------:|:----:|:------------:|:----:|:-------------:|
> | camoseg | 0.84 | **0.86** | 0.46 | **0.55** |
> | flood| 0.69 | **0.74** | 0.6 | **0.62** |
> | fiqa | 0.5 | **0.51** | 0.22 | **0.23** |
> | mldoc | 0.95 | **0.96** | 0.94 | **0.96** |
> | petfinder | **0.39** | 0.38 | 0.38 | **0.44** |
> | rvlcdip | 0.87 | 0.87 | **0.89** | 0.88 |
> | solar(10m)↓ | 1.49 | **0.38** | X | X |
> | yolanda↓ | **8.53** | 8.54 | 8.93 | **8.75** |
> | **Avg Token↓** | **63k** | 186k | **51k** | 146k |
> | **Avg Time↓** | **1904** | 5388 | **1731** | 4520 |
>
> **Key findings from our experiments:** Performance improvements are observed but are relatively modest, as the default setting already achieves strong results. For example, on the `camoseg` dataset, accuracy improves from 0.84 to 0.86, and on `flood` from 0.69 to 0.74. The improvement pattern is more pronounced in the challenging "-ext" setting (without external libraries), showing larger gains (e.g., `camoseg` from 0.46 → 0.55, `petfinder` from 0.38 → 0.44). **MLZero demonstrates clear capability to refine successful solutions, as optimizing an existing working solution is generally easier than achieving the first success.**
>
> However, our experiments reveal a trade-off: continuing iterations after the first success leads to performance improvements but comes with larger computational costs. Even when limiting to 5 iterations maximum, the token usage triples from 63k to 186k, and computation time increases from 1904s to 5388s in the default setting. This occurs because datasets that succeed in the first run will still complete all 5 iterations instead of stopping early. **Therefore, in the main paper, we chose our current "stop-at-first-success" design to prioritize efficiency while maintaining high performance.**
>
> We agree that an explicit iterative improvement is a promising direction for future work, especially in scenarios where computational resources are not a constraint. This would involve carefully balancing the trade-off between optimality and efficiency. **In the main paper, we also explored the system's potential in the 24hrs experiment (Table 1)**. By extending the time limit and using the best_quality preset for the generated code, MLZero's performance improves significantly, approaching expert-level human results on several datasets. This further validates our findings about the system's capability to produce higher-quality solutions when given more computational resources, while highlighting the practical value of our efficient default setting for most real-world applications.

---

> > ### Author Response · Authors · 2025-08-06
> >
> > We respectfully follow up regarding our rebuttal, in which we have addressed the concerns through additional experiments (on Continual Improvement of Solutions) and clarifications. As the discussion period concludes in two days, we would be grateful if you could review our response at your convenience. We remain available to address any further questions and sincerely appreciate your valuable input.

---

### Official Review · Reviewer_7xrS · 2025-07-01

**Clarity:** 3
**Significance:** 3
**Originality:** 3
**Rating:** 5
**Confidence:** 3

**Summary:**

The paper introduces MLZero, a novel multi-agent system that achieves fully automated machine learning (ML) across diverse data modalities with zero human intervention. MLZero integrates Large Language Models (LLMs) with a perception module and dual memory systems—semantic and episodic—to intelligently process raw multimodal data, iteratively generate and refine ML code, and deliver ready-to-use solutions. The system coordinates nine specialized agents to perform data understanding, library selection, code generation, execution, and error correction. MLZero demonstrates superior performance on both the MLE-Bench Lite and a newly proposed Multimodal AutoML Agent Benchmark, significantly outperforming existing approaches in success rate, efficiency, and robustness. Through ablation studies, the paper shows the crucial roles played by memory components in reducing hallucinations and improving code quality. This work presents a promising step toward general-purpose, autonomous ML agents.

**Questions:**

1. It would be better if authors are able to compare the token costs of your method.

2. Is the system robust to different models?

**Ethical Concerns:**

["NO or VERY MINOR ethics concerns only"]

**Limitations:**

See questions and weaknesses

**Quality:**

3

**Strengths And Weaknesses:**

Advantages:
1. Fully Automated Workflow: Handles the entire ML pipeline from raw data to results without human input.
2. Memory-Augmented Reasoning: Uses semantic and episodic memory to reduce hallucinations and improve reliability.
3. Strong Performance: Outperforms baselines with higher success rates and better solution quality.

Weakness:
1. Still lacks several experiments to validate the effectiveness of their method

---

> ### Author Rebuttal · Authors · 2025-07-29
>
> We thank you for the positive review and for highlighting the strengths of our fully automated workflow and memory-augmented reasoning.
>
> ---
>
> ## Q1: Comparison of Token Costs
>
> MLZero achieves superior performance (92% success rate) while using **4.3x fewer tokens than AIDE (63k vs. 273k per dataset) and costing only 23% as much (0.19 v.s. 0.82 \$/dataset), as shown in the table below.** Since most competitor systems don't report token usage, we estimated AIDE's consumption based on the cost of 25 benchmark runs. Codex CLI likely uses fewer tokens but at the significant cost of solution quality. This demonstrates MLZero's significant efficiency advantage while delivering better results.
>
> | System | Success Rate | Average Token Per Dataset | Average $Cost Per Dataset |
> |:-------------------------:|:------------:|:-------------------------:|:-------------------------:|
> | MLZero (def) | 92% | 63k | $0.19 |
> | MLZero (no suggested fix) | 75% | 51k | $0.15 |
> | AIDE (+ext) | 45.30% | ~273k | $0.82 |
>
> **Note:** MLZero (no suggested fix) was evaluated on 8 challenging datasets selected for ablation studies covering all modalities, explaining its lower success rate compared to the full 25-dataset benchmark used for MLZero (def) and AIDE (+ext). Despite using significantly fewer tokens (4.3x less than AIDE), MLZero achieves substantially better performance at a fraction of the cost.
>
> ---
>
> ## Q2: Is the system robust to different models?
>
> MLZero performs effectively across different LLM backbones. In Table 1 of the paper, we present results for **MLZero (def)**, which uses the large Claude 3.7 Sonnet model, and **MLZero (Llama 3.1 8B)**, which uses a much smaller model. Crucially, **MLZero (Llama 3.1 8B) still outperforms all other baseline agents**, including those using full-sized Claude 3.7 or GPT-4 class models, demonstrating **MLZero is agnostic to the choice of LLM backbones.**
>
> We also append here the performance of Claude Sonnet 3.7, **GPT 4.1**, and LLama 8B on 8 challenging datasets selected for ablation studies covering all modalities. Our results demonstrate that the MLZero framework is indeed robust to the choice of the underlying LLM.
>
> | Dataset | MLZero - Claude Sonnet 3.7 | MLZero - GPT 4.1 | MLZero - LLama 8B |
> |:------------------:|:--------------------------:|:----------------:|:-----------------:|
> | camoseg | 0.84 | 0.82 | X |
> | flood | 0.69 | 0.64 | 0.69 |
> | fiqa | 0.5 | 0.5 | 0.5 |
> | mldoc | 0.95 | 0.96 | 0.95 |
> | petfinder | 0.39 | 0.4 | 0.4 |
> | rvlcdip | 0.87 | 0.87 | X |
> | solar(10m)↓ | 1.49 | X | X |
> | yolanda↓ | 8.53 | 8.53 | 8.54 |

---

> > ### Comment · Reviewer_7xrS · 2025-08-05
> > **Keep Score Unchange as 5**
> >
> > After reading the rebuttal, I think 5 score fits this paper (Accept)

---

> > > ### Author Response · Authors · 2025-08-06
> > >
> > > Thank you for taking the time to review our rebuttal. We appreciate your thorough review and are glad our rebuttal addressed your concerns.

---

### Official Review · Reviewer_dDLh · 2025-07-04

**Clarity:** 3
**Significance:** 2
**Originality:** 3
**Rating:** 4
**Confidence:** 4

**Summary:**

This paper introduces MLZero, a novel multi-agent system that automates end-to-end machine learning workflows with minimal human intervention, specifically designed to handle raw, multimodal data. The system employs a perception module to interpret diverse inputs and uses dual semantic and episodic memory to improve iterative code generation while mitigating LLM hallucinations. When evaluated on the newly created Multimodal AutoML Agent Benchmark, MLZero achieved a 92% success rate, a 263.6% improvement over competitors. It also demonstrated superior performance on MLE-Bench Lite, securing six gold medals.

**Questions:**

please refer to the weakness part.

**Ethical Concerns:**

["NO or VERY MINOR ethics concerns only"]

**Final Justification:**

I'd like to thank the authors for their response. Some of my concerns have been addressed in the rebuttal. Therefore, I will keep my positve rating.

**Limitations:**

yes

**Quality:**

3

**Strengths And Weaknesses:**

Strengths
1.	Novel System Architecture: A well-designed multi-agent system featuring perception and dual-memory modules that automates the end-to-end ML workflow using complex, raw multimodal data.
2.	Rigorous and Extensive Evaluation: The paper introduces a new, challenging benchmark for raw data and provides comprehensive ablation studies and failure analyses to validate the system's effectiveness.
3.	State-of-the-Art Results: The system demonstrates impressive performance, achieving a 92% success rate (+263.6% improvement) and showing robust results even when using a much smaller 8B model.
4.	Clarity and Reproducibility: The work is clearly presented and highly reproducible, supported by a detailed appendix containing agent prompts, configurations, and comprehensive experimental results for verification.
Weaknesses & Question
1.	Novelty of Individual Components: While the system's integration is novel, its core memory modules (Semantic Memory akin to RAG , Episodic Memory to self-correction ) build on existing concepts. The paper should better contextualize this, clarifying the primary novelty lies in the successful application and synergy for the specific, complex domain of end-to-end AutoML rather than the invention of the components themselves.
2.	Scalability and Human Intervention: Performance relies on an offline, human-supervised process to create "condensed tutorials" for the knowledge base. This challenges the system's scalability and its "zero human intervention" claim. How much manual effort is required to integrate a new ML library, and how automated is this crucial knowledge preparation process?
3.	System Rigidity and Error Propagation: The system's linear, multi-agent pipeline creates a risk of cumulative error, while its effectiveness depends on a few high-level libraries. How does MLZero handle tasks outside its libraries' scope? More importantly, can a downstream coding agent detect and trigger a correction for an early-stage error, such as a flawed library selection made by the Perception module?

---

> ### Author Rebuttal · Authors · 2025-07-28
>
> We appreciate your positive assessment of our work, especially recognizing the novelty of the system architecture and the rigor of our evaluation. We address your specific concerns below.
>
> ---
>
> ## Q1: Novelty of Individual Components
>
> We agree that the foundational concepts behind our memory modules are established in the LLM literature. For semantic memory, we enable true lifelong learning capability. For episodic memory, we efficiently condense information to 400-500 characters per failed iteration (~150 tokens), which is sufficient for the limited number of iterations typically needed per task, allowing effective error correction without exceeding context limitations.
>
> **However, as you mentioned, the "successful application and synergy" is key. Our innovation lies in how we've adapted and implemented these concepts for the AutoML domain.** We believe the system innovation can be essential as other alternative approach greatly deteriorate the performance (Ablation study). Similarly, prompting semantic memory to other methods still can't outperform our framework (Table 1), which shows that synergy can be essential for AutoML agent. Furthermore, The effectiveness of this specific synergy is demonstrated by our complete elimination of API hallucination and perception errors (Table 3), which are primary failure modes for strong baselines like AIDE.
>
> **The novelty of MLZero, as a complete framework, has been mentioned in Line 55.** We will update it in the introduction as follows:
>
> > "Our primary contribution is not the invention of these components in isolation, but their novel, synergistic integration into a hierarchical multi-agent system specifically architected for the complex domain of end-to-end AutoML on raw, multimodal data".
>
> ---
>
> ## Q2: How much manual effort is required to integrate a new ML library, and how automated is this crucial knowledge preparation process?
>
> Adding a new ML library requires only about **one minute of human effort using our provided registration python script.** The user simply inputs four basic pieces of information: the library name (e.g., "FlagEmbedding"), version number (e.g., "FlagEmbedding==0.x.x"), a brief one-sentence description, and the path to the directory containing the library's official documentation. Users can optionally provide additional prompts for the library, but this is not required. Importantly, **no coding skills are needed** for this knowledge preparation process.
>
> After this brief human setup, our system takes over completely. The Summarization and Condensation agents (Appendix B.2) automatically process the documentation into structured, condensed formats to construct the Semantic Memory. While this processing takes time, it requires no further human intervention or supervision - the LLM agents handle all summarization, structuring, and knowledge extraction tasks autonomously.
>
> Once a library is integrated, MLZero operates from raw data to final prediction with no human input, as demonstrated in our experiments. This is a stark contrast to many existing systems that require manual preprocessing or hard-coded logic per dataset.
>
> ---
>
> ## Q3: How does MLZero handle tasks outside its libraries' scope?
>
> Our system is able to handle out-of-scope tasks effectively using a general fallback mechanism while selecting ML Libraries (outperforming all competitors even without specialized libraries). MLZero can fallback for tasks that fall outside the scope of its specialized, high-level libraries.
>
> **As detailed in Appendix D.5**, the system can select "machine learning" as a general tool. In this mode, the Coder agent is prompted to generate a solution from scratch using fundamental libraries (e.g., PyTorch, scikit-learn), leveraging the LLM's parametric knowledge. This provides flexibility and extends the system's applicability beyond the pre-registered libraries. Importantly, **our results show that even when all specialized ML libraries are removed (denoted as "-ext" in the main table), MLZero still achieves a rank of 4.94 and success rate of 69.3%, outperforming all competitors**. This demonstrates that our system remains effective even for tasks outside its specialized knowledge due to the effective system design.
>
>
> ---
>
> ## Q4: More importantly, can a downstream coding agent detect and trigger a correction for an early-stage error, such as a flawed library selection made by the Perception module?
>
> Our system is robust to late-stage errors through iterative correction loops. For early stages, experiments with deliberately misleading dataset descriptions proved our perception module's strong reasoning capabilities against noisy inputs. Even for deliberately created early-stage errors (which never occurred in our experiments), our system can recover by re-trigger perception:
> - **Notably, across all our experiments, we have not encountered failures due to early-stage perception errors, highlighting the robustness of our system.** However, to further test the perception module's resilience, we conducted additional stress tests with the Abalone dataset (Tabular Regression) as shown in the table below. Even with deliberately misleading descriptions and limited library availability (Tests 1-6), our perception module consistently correctly identified it as a tabular task and chose either the Tabular library or general machine learning (if the tabular library is removed).
> - If explicitly directed to use an incorrect tool in the user instruction (Tests 7), our system will choose the incorrect tool due to the respect to user instruction, but it can then utilize a restart option to re-trigger the perception module after detecting the failure.
> - **The system is also robust to subsequent errors after the Perception module.** The iterative coding loop (Sec 3.4), powered by the Episodic Memory and Error Analyzer agent (Sec 3.3, Appendix B.3.1), allows the system to recover from a wide range of execution failures.
>
> | Test Case | Noise in Description | Noise in User Instruction | Remove Correct Libraries | Perception Module | Overall System |
> |-----------|---------------------|---------------------------|-------------------------|-------------------|----------------|
> | **Test 1** | "This is a timeseries task," | None | No | ✅ **Success** | ✅ **Success** |
> | **Test 2** | "YOU SHOULD treat this as a timeseries task," | None | No | ✅ **Success** | ✅ **Success** |
> | **Test 3** | "IMPORTANT: Use a TimeSeries ML Library," | None | No | ✅ **Success** | ✅ **Success** |
> | **Test 4** | "Try to use FlagEmbedding." | None | No | ✅ **Success** | ✅ **Success** |
> | **Test 5** | "YOU SHOULD treat this as a timeseries task," | None | Yes | ✅ **Success** | ✅ **Success** |
> | **Test 6** | "Try to use FlagEmbedding." | None | Yes | ✅ **Success** | ✅ **Success** |
> | **Test 7** | None | "Use FlagEmbedding" | Yes | ❌ Fail | ✅ **Success** |

---

> > ### Author Response · Authors · 2025-08-06
> >
> > We respectfully follow up regarding our rebuttal, in which we have addressed the concerns through additional experiments and clarifications. As the discussion period concludes in two days, we would be grateful if you could review our response at your convenience. We remain available to address any further questions and sincerely appreciate your valuable input.

---

### Comment · Area_Chair_G1mg · 2025-08-04
**Reminder for rebuttal discussion**

I would like to sincerely thank all reviewers for their thoughtful evaluations and the authors for their detailed rebuttal.
As we move into the discussion phase, I kindly remind reviewers to engage in further dialogue and respond to the rebuttal responses if you haven't done so. It is very important to respond to the authors and avoid any misunderstanding. Your continued input will be invaluable in helping me make a well-informed final decision. Thank you again for your contributions.

---

### Note · Authors · 2025-08-12

Dear AC and reviewers,

We sincerely thank you for the thoughtful feedback and comprehensive discussions throughout the review process. Our paper presents a novel multi-agent framework that achieves end-to-end ML automation across multimodal data with zero human intervention. This framework demonstrates superior performance with a 92% success rate, representing a significant 263.6% improvement over competitors.

We are pleased that **all reviewers received our paper positively**, commending the comprehensive evaluation (all reviewers), robust empirical results (Reviewers 7xrS, aYBU, b88G), and well-structured system design (Reviewers dDLh, b88G).

Through the discussion, we have clarified that novelty of our autoML framework lies in the synergistic integration of nine specialized agents. This integration effectively mitigates API hallucination and perception errors. We also presented extensive additional experiments demonstrating robustness to both the choice of LLMs and deliberately misleading inputs. Furthermore, we provided a detailed token efficiency analysis and highlighted our simplified library integration process. These results collectively demonstrate that MLzero is **a robust, efficient and effective AutoML system**.

We are grateful for the rigorous review process and would greatly appreciate your support.

Best,

The Authors

---

### Decision · Program_Chairs · 2025-09-17

**Decision:**

Accept (poster)

**Comment:**

(a) Summary of scientific claims and findings
This paper presents MLZero, a hierarchical multi-agent framework for end-to-end ML automation across multimodal data. The system integrates a perception module with semantic and episodic memory to reduce hallucinations and improve iterative code generation. MLZero is evaluated on MLE-Bench Lite and a newly introduced Multimodal AutoML Agent Benchmark (25 diverse tasks), achieving a high success rate and outperforming strong baselines. The framework also demonstrates efficiency (fewer tokens, lower cost) and robustness across different LLM backbones, including smaller models.

(b) Strengths
- Well-structured system design with nine coordinated agents.
- Strong empirical results: consistent improvements over competitive baselines on two benchmarks.
- Comprehensive evaluation
- Clear and reproducible presentation

(c) Weaknesses
- Limited novelty in individual components; many techniques are adaptations of known methods.
- Heavy emphasis on engineering, lacks deeper insights.
- The newly introduced benchmark is authored by the same team and has not been externally validated.
- Minimal exploration of LLM limitations and failure cases beyond quantitative error rates

(d) Reasons for decision
The paper makes a solid systems contribution: while the novelty lies more in integration than in new algorithms, the framework is thoughtfully designed, rigorously evaluated, and demonstrates clear performance gains. The authors' rebuttal convincingly addressed concerns about scalability, benchmark fairness, and robustness. The work is impactful for the AutoML and agent communities, but the lack of fundamental theoretical advances or stronger novelty makes it better suited as a poster rather than spotlight/oral.

(e) Discussion and rebuttal period
Reviewers raised questions on novelty, benchmark design, scalability, and system robustness. The authors clarified that adding new libraries requires minimal human effort, provided additional results on robustness across LLMs, offered token/cost comparisons, and presented stress tests and error analyses. While concerns on originality and benchmark neutrality remain partly valid, the rebuttal addressed most technical doubts satisfactorily. Overall, the reviewers converged on a positive but cautious stance, leaning toward borderline accept. I weigh the strong empirical contribution and practical relevance over the novelty concerns, leading to my recommendation for poster acceptance.